# The C-terminus of the prototypical M2 muscarinic receptor localizes to the mitochondria and regulates cell respiration under stress conditions

Irene Fasciani[1◉], Francesco Petragnano[1◉], Ziming Wang[2], Ruairidh Edwards[3], Narasimha Telugu[2], Ilaria Pietrantoni[1], Ulrike Zabel[4], Henrik Zauber[2], Marlies Grieben[2], Maria E. Terzenidou[3], Jacopo Di Gregorio[1], Cristina Pellegrini[1], Silvano Santini, Jr[5], Anna R. Taddei[6], Bärbel Pohl[2], Stefano Aringhieri[7], Marco Carli[7], Gabriella Aloisi[1], Francesco Marampon[8], Eve Charlesworth[9], Alexandra Roman[2], Sebastian Diecke[2], Vincenzo Flati[1], Franco Giorgi[7], Fernanda Amicarelli[5], Andrew B. Tobin[3], Marco Scarselli[7], Kostas Tokatlidis[3], Mario Rossi[1], Martin J. Lohse [2,4,10]*, Paolo Annibale [2,9]*, Roberto Maggio[1]*

1 Department of Biotechnological and Applied Clinical Sciences, University of L'Aquila, L'Aquila, Italy, 2 Max Delbrück Center for Molecular Medicine, Berlin, Germany, 3 Centre for Translational Pharmacology, Institute of Molecular, Cell and Systems Biology, College of Medical, Veterinary and Life Sciences, University of Glasgow, Glasgow, United Kingdom, 4 Institute of Pharmacology and Toxicology, University of Würzburg, Würzburg, Germany, 5 Department of Life, Health and Environmental Sciences, University of L'Aquila, L'Aquila, Italy, 6 Section of Electron Microscopy, Great Equipment Center, University of Tuscia, Viterbo, Italy, 7 Department of Translational Research and New Technology in Medicine, University of Pisa, Pisa, Italy, 8 Department of Radiotherapy, University of Roma La Sapienza, Roma, Italy, 9 School of Physics and Astronomy, University of St Andrews, St Andrews, United Kingdom, 10 ISAR Bioscience Institute, Munich, Germany

◉ These authors contributed equally to this work.
* martin.lohse@isarbioscience.de (MJL); pa53@st-andrews.ac.uk (PA); roberto.maggio@univaq.it (RM)

**Data Availability Statement:** Source data related to the Figures and Supplementary Figures in this manuscript can be found archived by Figure in 'S1

## Abstract

Muscarinic acetylcholine receptors are prototypical G protein-coupled receptors (GPCRs), members of a large family of 7 transmembrane receptors mediating a wide variety of extracellular signals. We show here, in cultured cells and in a murine model, that the carboxyl terminal fragment of the muscarinic $M_2$ receptor, comprising the transmembrane regions 6 and 7 ($M_2$tail), is expressed by virtue of an internal ribosome entry site localized in the third intracellular loop. Single-cell imaging and import in isolated yeast mitochondria reveals that $M_2$tail, whose expression is up-regulated in cells undergoing integrated stress response, does not follow the normal route to the plasma membrane, but is almost exclusively sorted to the mitochondria inner membrane: here, it controls oxygen consumption, cell proliferation, and the formation of reactive oxygen species (ROS) by reducing oxidative phosphorylation. Crispr/Cas9 editing of the key methionine where cap-independent translation begins in human-induced pluripotent stem cells (hiPSCs), reveals the physiological role of this process in influencing cell proliferation and oxygen consumption at the endogenous level. The expression of the C-terminal domain of a GPCR, capable of regulating mitochondrial function, constitutes a hitherto unknown mechanism notably unrelated to its canonical signaling function as a GPCR at the plasma membrane. This work thus highlights a potential novel

Data'. Source files for the microscopy images displayed in the Figures and Supplementary Figures in this manuscript can be found on FigShare: 10.6084/m9.figshare.25249912.

**Funding:** This project was funded by the Deutsche Forschungsgemeinschaft (DFG, German Research Foundation) through Project 421152132 SFB1423 subproject C03 (PA and MJL) (https://gepris.dfg. de/gepris/projekt/431599318). This study was further supported by European Union's Horizon2020 Marie Skłodowska-Curie Actions (MSCA) Program under Grant Agreements 641833 and 860229 (ONCORNET and ONCORNET2.0 to MJL (https://oncornet.eu). We are grateful for funding to the European Union - NextGenerationEU under the Italian Ministry of University and Research (MUR) National Innovation Ecosystem grant ECS00000041 - VITALITY - CUP E13C22001060006 (IF) (https://next-generation-eu.europa.eu). PA would like to gratefully acknowledge support from the Leverhulme Trust (RL-2022-015) (www.leverhulme.ac.uk). MR would like to acknowledge funding from University of L'Aquila through project 07_PROGETTO_RICERCA_ATENEO_ Rossi (https://www.univaq.it). Work in the KT lab was supported by grants UK Research and Innovation-Biotechnology and Biological Sciences Research Council (UKRI-BBSRC) BB/T003804/1, BB/ R009031/1, BB/X511948/1 and UKRI-Medical Research Council (UKRI-MRC) MC_PC_19039 (https://www.ukri.org). IP acknowledges a short term exchange grant in 2017 from the Deutscher Akademischer Austauschdienst (DAAD) (http:// www.daad.de). FP Acknowledges support from the University of L'Aquila towards international mobility in 2020 (https://www.univaq.it). JDG was supported by Fondazione Umberto Veronesi (https://www.fondazioneveronesi.it). MJL would like to gratefully acknowledge support from the Bavarian Ministry of Economics (ISAR Bioscience Institute) (https://www.stmwi.bayern.de). The funders did not play any role in the study design, data collection and analysis, decision to publish, or preparation of the manuscript.

**Competing interests:** The authors have declared that no competing interests exist.

**Abbreviations:** CNV, copy number variation; DMEM, Dulbecco's Modified Eagle Medium; ETC, electron transport chain; FBS, fetal bovine serum; FCS, fluorescence correlation spectroscopy; FRET, Fluorescence Resonance Energy Transfer; GPCR, G protein-coupled receptor; hiPSC, human-induced pluripotent stem cell; i3 loop, 3rd intracellular loop; IEM, immunoelectron microscopy; IRES, internal ribosome entry site; lncRNA, long non-coding

mechanism that cells may use for controlling their metabolism under variable environmental conditions, notably as a negative regulator of cell respiration.

## Introduction

Muscarinic receptors include 5 structurally related receptors, belonging to class A of the G protein-coupled receptor (GPCR) family [1]. They are comprised of 7 transmembrane domains (TM domains 1–7) connected by intra and extracellular loops. Of these structural elements, the third intracellular (i3) domain is the one conferring a coupling specificity for G proteins, β-arrestin recruitment, and endocytic internalization [2]. However, only a small portion of the i3 loop is required for receptor function, and for this reason, large deletions of this segment have no effect on receptor activation and internalization [3].

GPCRs are normally distributed at the plasma membrane, where, after activation by an agonist and coupling to either G proteins or β-arrestin, regulate multiple intracellular effectors. Nevertheless, non-canonical localizations of GPCRs have also been described, such as in endosomes [4], cell division compartments (centrosomes, spindle midzone, and midbodies) [5], and mitochondria [6,7], underpinning nontraditional roles for this protein family.

Several studies have shown that the GPCR structurally related transmembrane protein bacteriorhodopsin consists of autonomous folding domains that translocate to the membrane and interact with each other to function as a proton pump [8]. Interestingly, it was also shown that fragments of muscarinic receptors can interact to form functionally active chimeric receptors, highlighting the conserved evolutionary ability of domains from these receptor family to interact among each other, as well as with other proteins [9]. Intriguingly, in 2016, we observed that the insertion of a stop codon in the third intracellular loop of the muscarinic $M_2$ receptor was not able to abolish its function [10], presenting the suggestive possibility that a cap-independent translation of the $M_2$ C-terminal domain may occur in this mutant, i.e., that the $M_2$ mRNA may lead to the translation not only of the full-length receptor, but also of its C-terminal domain, that we name here $M_2$tail.

In this work, we therefore set out to explore whether the expression of the $M_2$tail, a region containing the TM regions 6 and 7, occurs naturally by a cap-independent translation mechanism. Combining radioligand binding, molecular biology approaches, advanced fluorescence imaging, and Crispr/Cas9 genetic editing of human-induced pluripotent stem cells (hiPSCs), we have gathered compelling data indicating a novel translation mechanism for C-terminal domain of the $M_2$ muscarinic receptor. This translation mechanism appears to be regulated by an internal ribosome entry site (IRES) sequence within the $M_2$ mRNA allowing the independent expression of the receptor's C-terminal domain as a distinct polypeptide.

Moreover, by using bimolecular fluorescence complementation, in vitro mitochondrial import, and cell metabolism assays, we observe that $M_2$tail is strongly targeted to the cell mitochondria. Importantly, this mitochondrial targeting has functional ramifications since both cap-dependent mediated overexpression of $M_2$tail, as well as cap-independent mediated translation starting from the mRNA of the full-length $M_2$ receptor, resulted in a sizable reduction of oxidative phosphorylation. In fact, we show that cellular stress favors the activation of this newly discovered IRES-dependent translation of $M_2$tail and that the fragment interacts with the Complex V of the respiratory chain on the mitochondria inner membrane. The implication of this newly discovered regulatory mechanism of the IRES-mediated translation of the $M_2$ C-terminal domain might have profound physiological and pathophysiological implications.

RNA; OCR, oxygen consumption rate; PBS, phosphate-buffered saline; PCC, Pearson correlation coefficient; ROS, reactive oxygen species; SBTI, soybean trypsin inhibitor; sgRNA, small guide RNA; smORF, small open reading frame.

In summary, our study reports the new observation that alternative reading of a GPCR receptor gene can translate into a fragment of the receptor that serves a cellular function entirely different from canonical GPCR signaling. Taken together, these data strongly suggest that the $M_2$ mRNA yields not only a plasma membrane receptor able to transduce many of the acetylcholine functions but also $M_2$tail, a protein fragment that reduces the activity of mitochondria and reactive oxygen species (ROS) accumulation and it might represent a hitherto unappreciated step for in vivo anti-oxidative stress, during for instance reperfusion, tissue damage, and disease-promoted metabolic cellular stress.

## Results

### Cap-independent production of the C-terminus of the $M_2$ receptor and evidence for an IRES

We began our work by testing our earlier hypothesis that the last 2 TM regions (TM6-7) of $M_2$stop228, an $M_2$ muscarinic receptor mutant bearing a stop codon at the beginning of the i3 loop in position 228 (**S1 Fig**), may still be translated [10]. We thus generated EGFP C-terminal fusion of the wild-type $M_2$R ($M_2$wt) receptor and $M_2$stop228: **Fig 1A** shows that cells expressing $M_2$stop228-EGFP display a green fluorescence signal. Moreover, the signal displays a predominant intracellular localization with respect to the $M_2$wt, which instead localizes—as expected—largely at the plasma membrane. A key observation is that the specific location of the stop codon upstream of the i3 loop appears to have particular relevance. For example, if a stop is instead inserted downstream of position 228, e.g., at positions 400, right before the first methionine in TM6, no fluorescence can be observed in cells expressing the mutant $M_2$stop400-EGFP **Fig 1A.**

To further appreciate the origin of this unexpected process—the 228 stop codon should in principle abolish any EGFP signal—we performed radioligand binding experiments in cells transfected with this mutant. $M_2$stop228 exhibited $K_D$ value for the antagonist [$^3$H]NMS (N-methylscopolamine) calculated from direct saturation experiments, as well as $IC_{50}$ values, obtained in competition experiments for the agonist carbachol, similar to those calculated for the $M_2$wt (**Table 1**), albeit displaying a very low level of binding sites at the plasma membrane, specifically 45 fmol/mg of protein. Furthermore, $M_2$stop228 was found functionally active as it was capable to inhibit adenylyl cyclase [10] and increase ERK phosphorylation (**S2A Fig**), even though the extent of response was reduced if compared to $M_2$wt (**Table 1**). As a control, receptor fragments, translated from mRNAs comprising only the sequence for TM1-5 (i.e., whose sequences run only until position 228 or 283 of the corresponding full-length receptor), called $M_2$trunk(1–228) or $M_2$trunk(1–283), did not show any radioligand binding activity, despite coding for the same first 5 trans-membrane regions (TM1-5) that the canonical translation of $M_2$stop228 should also produce (**Fig 1B** and **Table 1**) [9].

Notably, our previous research has demonstrated the ability of distinct fragments of muscarinic receptors, cotransfected together, to reassemble into fully functional receptors. Specifically, we showed that the last 2 TMs (TM6-7) of the $M_2$ receptor can combine with TM1-5 to recreate a functional receptor [9] (**Table 1**), supporting the notion that the explanation of the apparent paradox of $M_2$stop228 lies in the independent translation of $M_2$tail as a distinct polypeptide, as graphically summarized in **Fig 1B**.

### Evidence of an IRES-mediated $M_2$ C-terminus translation

$M_2$stop228 mRNA appears thus able to give rise—despite the presence of a stop codon in the i3 loop—to 2 distinct protein products (a *trunk* TM1-5 and a *tail* TM6-7), in turn able to reconstitute a functional receptor at the plasma level.

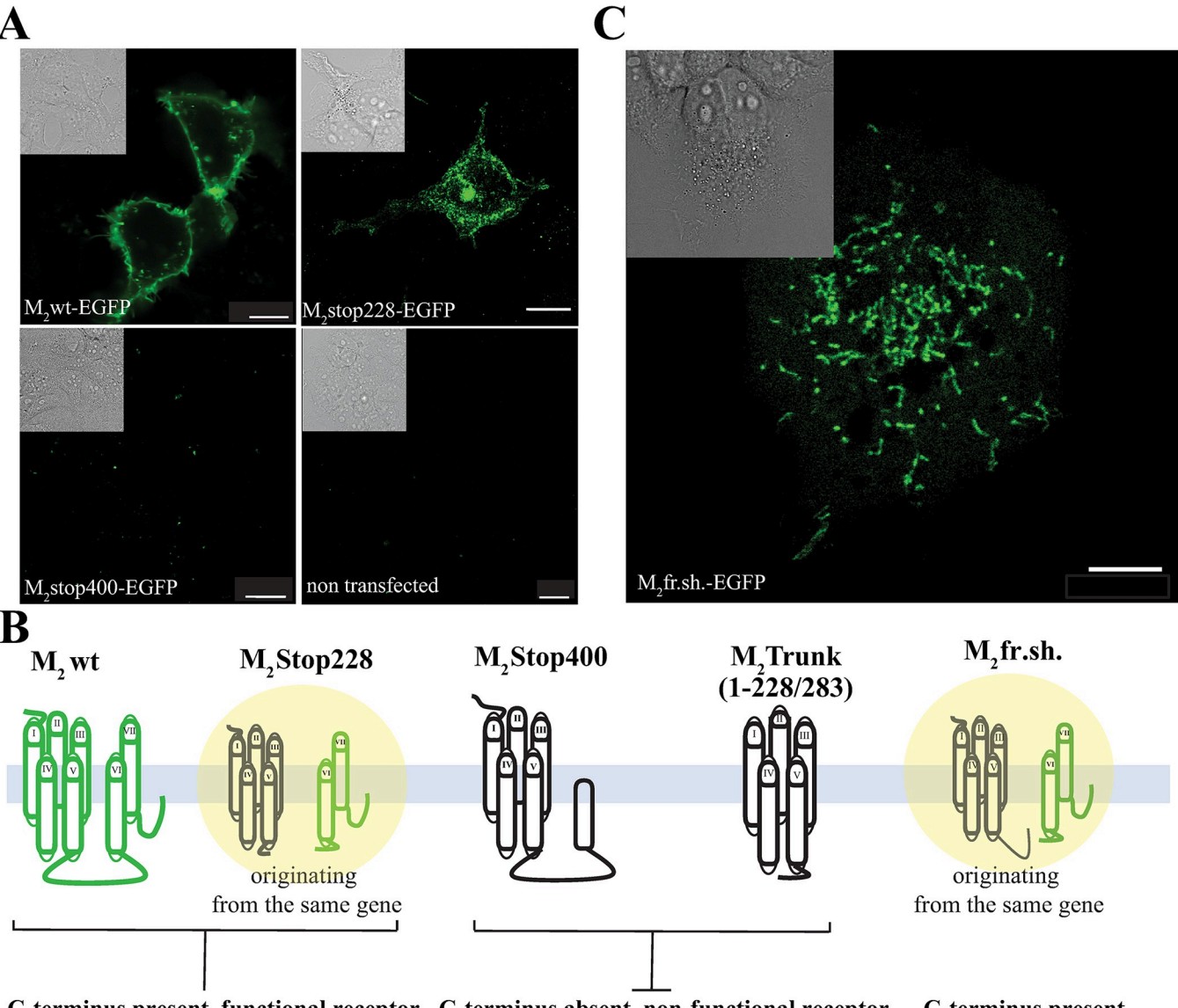

**Fig 1. The c-terminus of the M$_2$ receptor is expressed by a cap-independent translation mechanism, originating in the i3 loop.** (**A**) Fluorescence micrographs of HEK293 and COS-7 cells respectively expressing M$_2$-EGFP, M$_2$stop228-EGFP, M$_2$stop400-EGFP, together with an untransfected control. (**B**) Schematic representation of the gene products that reach the plasma membrane of the M$_2$wt, M$_2$stop228, M$_2$trunk(1–283), and M$_2$fr.sh. (**S3 Fig**). (**C**) Fluorescence micrograph of a representative HEK293 cell expressing M$_2$fr.sh.-EGFP. Insets display DIC image of the same field of view (scale bar 10 μm). The imaging settings remain consistent for fluorescence images, but contrast adjustments were made to enhance visualization. In the untransfected cells and M$_2$stop400 panels, the contrast was set to 0. Insets display DIC image of the same field of view (scale bars 10 μm).

To challenge this hypothesis and gain further insight into the mechanism at play, we generated a frame shift mutant of the M$_2$ receptor: we inserted a single guanine nucleotide in the sequence of the M$_2$-EGFP, changing the methionine in position 368 into an aspartate, which in turn generates an in-frame stop codon at position 370 (**S1 and S3 Figs**). The observation that cells transfected with M$_2$fr.sh.-EGFP retain green fluorescence is most remarkable (**Fig 1C**), since the EGFP is now out of the original reading frame and it can only be properly translated by a cap-independent translation mechanism, starting from a methionine in the new reading frame.

**Table 1. Radioligand binding and activation properties of $M_2$ receptor mutants.**

| Receptor | Binding data | | | Adenylyl cyclase inhibition assay | |
|---|---|---|---|---|---|
| | $B_{max}$ (fmol/mg) | [3H]NMS $K_D$ (pM) | Carbachol IC$_{50}$ (μM) | Carbachol EC$_{50}$ (μM) | Maximum inhibition of forskolin stimulated cAMP accumulation (%) |
| $M_2$ | 980 ± 60 | 62 ± 7 | 2.1 ± 0.6 | 0.04 ± 0.01 | 53 ± 5 |
| $M_2$trunk(1–283) | N.B. | | | | |
| $M_2$trunk(1–228) | N.B. | | | | |
| $M_2$stop228 | 45 ± 2 | 70 ± 10 | 1.8 ± 0.4 | 0.13 ± 0.02 | 27 ± 3 |
| $M_2$trunk(1–283) +$M_2$tail(M-281-466) | 480 ± 20 | 79 ± 10 | 0.7 ± 0.2 | 0.15 ± 0.03 | 37 ± 4 |
| $M_2$trunk(1–228) +$M_2$tail(368–466) | 70 ± 10 | 72 ± 8 | 0.6 ± 0.2 | 0.17 ± 0.03 | 29 ± 4 |

Binding properties and inhibition of forskolin stimulated adenylyl cyclase by $M_2$, $M_2$stop228, $M_2$trunk(1–228), $M_2$trunk(1–283), and cotransfected $M_2$trunk(1–283) + $M_2$tail(M-281-466) and $M_2$trunk(1–228) + $M_2$tail(368–466). Untransfected COS-7 cells did not show any [3H]NMS-specific binding. N.B. = no specific [3H]NMS binding. Data are averages with SEM.

Furthermore, the observation that both $M_2$stop228-EGFP and $M_2$fr.sh.-EGFP retain green fluorescence, whereas $M_2$stop400-EGFP does not, indicates not only that a cap-independent translation mechanism occurs, but also that the starting methionine for the process must be downstream of stop 228 and upstream of stop 400, leaving as a leading candidate one of the in-frame methionines of the i3 loop.

To narrow this down, we employed radioligand binding data from expression of receptor mutants carrying a double stop codon, the first in position 228 and the second in positions ranging from 248 to 400, including mutations of the first (M248) and second (M296) in-frame methionines in the third i3 loop (S1 Table and S3 Fig). The observation that mutation of the first 2 methionines does not affect $B_{max}$ points to M368 as the most likely site for the cap-independent initiation of translation of the $M_2$tail, that we now term more specifically $M_2$tail(368–466). These binding results can furthermore rule out read-through and termination re-initiation: if these mechanisms were responsible for translation of the C-terminal of the $M_2$stop228, suppression of the first methionine (M248) would have abolished the binding. We shall further note that cells expressing $M_2$stop228 showed only 1 single $M_2$-specific cDNA band supporting the concept that the $M_2$tail(368–466) was produced from the same $M_2$-mRNA molecule by a cap-independent mechanism instead of, for example, by RNA splicing variants (S2B Fig and S1 Methods). However, upon immunoblotting the protein products it is immediate to observe that the full-length $M_2$ receptor produces a set of low molecular weight band matching those observed for $M_2$tail(368–466), when the constructs possess a C-terminal tag (myc) (S2C Fig). As depicted in the graphical representation in S2C Fig, the $M_2$wt exhibits a band pattern comprising 3 high molecular weight bands corresponding to the unmodified receptor (55.3 kDa). Additionally, there are 2 supplementary bands indicating glycosylation of the receptor [11]. The low molecular weight bands resembling those found in the $M_2$tail(368–466) suggest the presence of the naked $M_2$tail(368–466) fragment, measuring approximately 14.9 kDa, alongside a form at approximately 23 kDa, likely indicative of a posttranslational modification such as monoubiquitination. This aligns with the molecular weight of ubiquitin addition, approximately 8.6 kDa [12]. In contrast, the N-terminally tagged $M_2$stop228 produces a band in line with that observed for cells transfected with the $M_2$trunk(1–228) fragment (S2D Fig), further ruling out stop-codon read-through, as in this instance, we should have observed a band pattern corresponding to the full-length $M_2$wt (S2D Fig). Comparable results are observed in COS-7 cells (S2E Fig).

## IRES position within the $M_2$ coding sequence

To further validate the results obtained working with full-length $M_2$ receptor mutants and to investigate the critical nucleotides for IRES activity, we generated a fluorescence reporter construct (Sirius-$M_2$i3(417n)-EGFP) in which the i3 loop of $M_2$ was inserted between the 2 fluorescent proteins, Sirius and EGFP (Fig 2A). The sequence for Sirius is followed by a stop codon, which means that Sirius-$M_2$i3(417n)-EGFP translates Sirius according to the classical cap-dependent mechanism, serving as an indicator of transfection efficiency, and, at the same time, drives EGFP translation only if the $M_2$i3 loop contains an IRES sequence, essentially functioning as a bicistronic expression cassette. We transfected several cell lines with this construct, and all of them were positive for the blue and green fluorescence (S4A Fig). HEK293 cells, used throughout our study, appeared positive for both the blue and green fluorescence, as shown in the inset of Fig 2A and in line with the results reported in Fig 1A. To further rule out the possibility that EGFP translation from the Sirius-$M_2$i3(417n)-EGFP construct could be

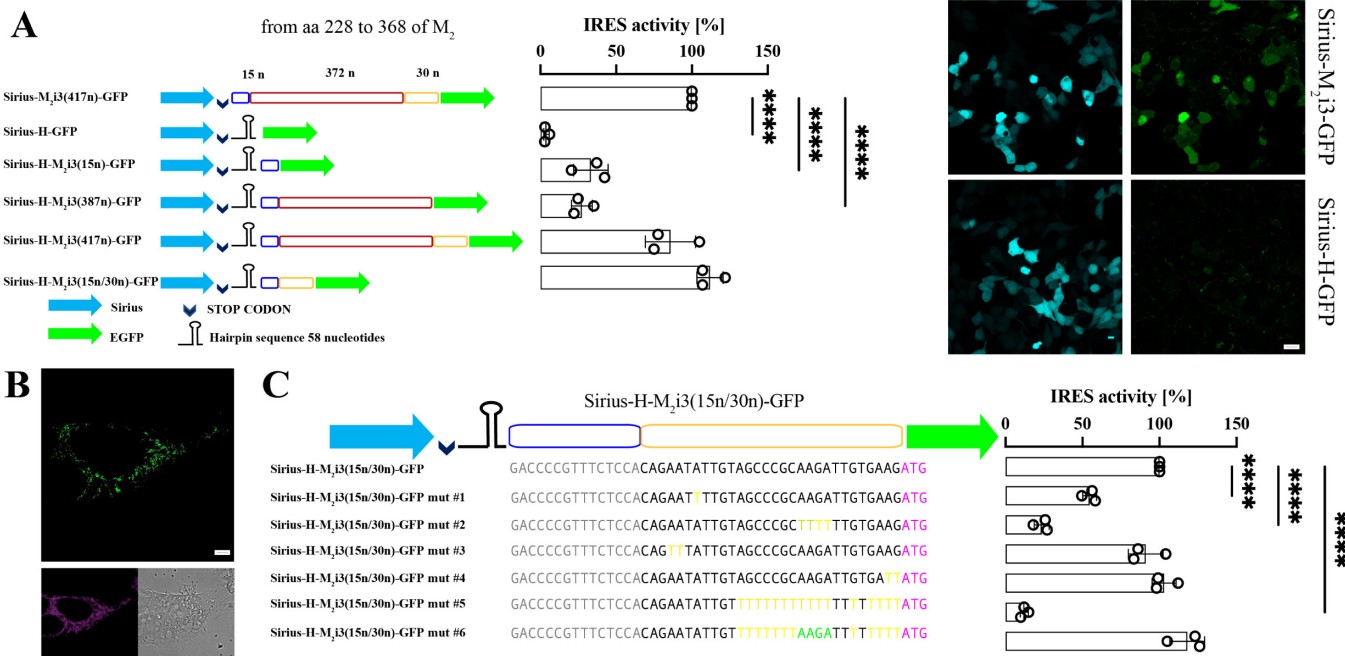

**Fig 2. Localization of the IRES within the i3 loop the muscarinic $M_2$ receptor using a fluorescence reporter assay.** (**A**) **Left:** Schematic representation of bicistronic plasmids bearing the $M_2$ i3 loop between the coding regions of the Sirius (blue arrow) and EGFP (green arrow) fluorescent proteins. A guanine nucleotide was originally inserted before the EGFP in the plasmid Sirius-H-EGFP to maintain in frame the sequences of the 2 fluorescent proteins. In all the other plasmids, the sequence of the 2 fluorescent proteins is not in frame. Fifteen nucleotides (blue sequence) derived from the $M_2$ i3 loop were inserted between the end of Sirius and the hairpin loop, to allow the easy scanning of the ribosome across the stop codon. The red and yellow sequences represent the segment of 402 nucleotides (372 +30) of the i3 loop. The codon for M368A is immediately after the 30 nucleotides represented by the yellow sequence. **Right:** Efficiency of cap-independent translation for the control Sirius-$M_2$i3(417n)-EGFP and the 6 mutants. Note that in these bicistronic constructs, the M368 corresponds to the initiation of the EGFP protein, as denoted by the green arrow. The blue, red, and yellow sequences represent segments of the M2 i3loop from aa 228 to 368. The insert shows the difference in the green fluorescence between 2 of the mutants. Significance values reported in the graphs were determined by a one-tailed Student $t$ test, $p$-values: ** $0.001 < p < 0.01$. (**B**) Expression of the construct $M_2$-Hairpin-EGFP in HEK293 cells: (top) Confocal images of $M_2$-Hairpin-EGFP, Mitotracker Deep red staining and DIC (bottom). Scale bar 5 μm. (**C**) **Left:** Mutations of the bicistronic Sirius-H-$M_2$i3(15n/30n)-EGFP plasmids in the region of the $M_2$ i3 loop from nucleotides 1972 to 2001, in yellow are the modified nucleotides. In purple the codon for M368. In Sirius-H-$M_2$i3(15n/30n)mut#6-EGFP the 4 nucleotides in green (AAGA) were reinserted in the stretch of 20 T of Sirius-H-$M_2$i3(15n/30n)mut#5-EGFP. **Right:** IRES activity, highlighting the efficiency of cap-independent translation for the control Sirius-H-$M_2$i3(15n/30n)-EGFP and the 6 mutants illustrated. Significance ($p < 0.01$) was calculated in **A** against Sirius-$M_2$i3(417n)-EGFP and in **B** against Sirius-H- $M_2$i3(15n/30n)-EGFP. Error bars are SEM. Significance values reported in the graphs were determined by a one-tailed Student $t$ test, $p$-values: ** $0.001 < p < 0.01$. Source data for panels A and C can be found in S1 Data. IRES, internal ribosome entry site.

attributable to read-through and termination-re-initiation mechanisms, an mRNA hairpin loop was inserted in the sequence after the stop codon of the Sirius protein and before the $M_2$i3(417n), resulting in Sirius-H-$M_2$i3(417n)-EGFP. This modification did not eliminate the EGFP signal, showing only a slight reduction of the EGFP/Sirius with respect to the value measured for the analogous reporter without hairpin (approximately 90%) (**Fig 2A**), underscoring that read-through and termination-re-initiation mechanisms do not appear to be at play. In contrast, when the $M_2$i3(417n) segment was removed from Sirius-H-$M_2$i3(417n)-EGFP, yielding Sirius-H-EGFP, it entirely abolished EGFP expression (**Fig 2A**). This observation indicates that the hairpin loop effectively blocks read-through and termination-re-initiation mechanisms.

To narrow down the position of the IRES sequence, we divided the $M_2$ i3 loop in 3 parts and the resulting nucleotide sequences, 685–699, 685–1071, and 685-699/1072-1101, were inserted downstream of the hairpin loop in 3 different constructs (**Figs 2A** and **S3**).

Only when the last portion of the $M_2$ i3 loop was inserted in the construct after the hairpin, leading to Sirius-H-$M_2$i3(15n/30n)-EGFP, approximately 110% of the value measured for the Sirius-$M_2$i3(417n)-EGFP plasmid was recovered (**Fig 2A**). Notably, when the hairpin was inserted within the i3 loop of the full-length $M_2$-EGFP, upstream of the M368, as in Sirius-H-$M_2$i3(417n)-EGFP, it was still possible to observe green fluorescence in transfected cells (**Fig 2B**).

These results confirm that the IRES is located in a region of the i3 loop sequence immediately upstream of the third methionine (M368). This is also confirmed from ribosome profiling data from human left ventricle tissue, displaying increased P-sites as well as overall ribosomal coverage in correspondence of this region (**S4B Fig**). Based on this evidence, we mutated the conserved residues highlighted in yellow in **S4C Fig,** in order to check for their relevance in recruiting the ribosome on the $M_2$ i3 loop sequence, as shown in **Fig 2C**. We found that the 4 nucleotides AAGA1090-1093 next to the methionine 368 play a pivotal role in recruiting the ribosome on the $M_2$ mRNA and in translating the $M_2$ C-terminal fragment, as discussed more in detail in **S1 Text**.

## Generalization to other members of the GPCR superfamily

In order to determine if the observed behavior is specific to the $M_2$ receptor or could highlight a feature common to other members of the GPCR superfamily, we repeated several of our experiments in another GPCR, the related muscarinic $M_3$ receptor. The $M_3$ displays an in-frame methionine at the end of the i3loop, in position 481. We then inserted a stop codon in position 273, generating the construct $M_3$stop273 that still displayed radioligand binding and functional activity (**S2 Table**). Several other mutants, mimicking the single and double stop mutations made for the $M_2$ displayed a behavior consistent with M481 being the start point of a cap-independent translation of the C-terminal also for the $M_3$ receptor (**S2 Table**), thus indicating that IRES-mediated cap-independent translation of the C-terminal domain may not be an exclusive phenomenon confined solely to the $M_2$ receptor, but may be a shared characteristic among some other members of the GPCR superfamily. The presence of an IRES sequence upstream of M481 further appears to be supported by the observation of GFP fluorescence when 558 nucleotides from the i3 loop of the $M_3$ receptor (from nucleotide 880 to 1437) were inserted into the identical Sirius-EGFP reporter construct used for the $M_2$ receptor (**S5A and S5B Fig**). Furthermore, the sequence alignment of the last portion of the $M_2$ i3 loop (from nucleotide 1072 to 1104) with regions adjacent to the second ($M_1$), fourth ($M_3$ and $M_4$), and first ($M_5$) in-frame methionines, clearly shows that some nucleotide residues are conserved (**S4C Fig**). Overall, these findings underline that cap-independent translation could be a more general feature of GPCRs mRNAs.

## Mitochondrial localization of the $M_2$ C-terminal fragment generated by cap-dependent expression under the control of a viral promoter

Having established that the expression of the $M_2$ C-terminal domain is regulated by an IRES and that its translation starts with M368, we proceeded to investigate the role it might play in the cell. For this purpose, we fused $M_2$tail(368–466) with EGFP in pcDNA3, that is under the control of a strong CMV promoter, and we expressed it in HEK293 cells in order to study its subcellular localization. Confocal microscopy revealed a remarkable mitochondrial localization of the $M_2$tail(368–466)-EGFP construct (**Fig 3A**).

We confirmed this observation by co-staining HEK293 cells with a conventional mitochondrial stain and by co-expressing $M_2$tail(368–466)-EGFP with Tom20-mCherry, a marker for the outer mitochondrial membrane (**S6A Fig**). The observed mitochondrial localization of $M_2$tail(368–466)-EGFP appeared in stark contrast to the canonical cellular localization of the $M_2$ receptor, as shown in **Fig 3B**, that displays confocal sections of cells co-transfected with the

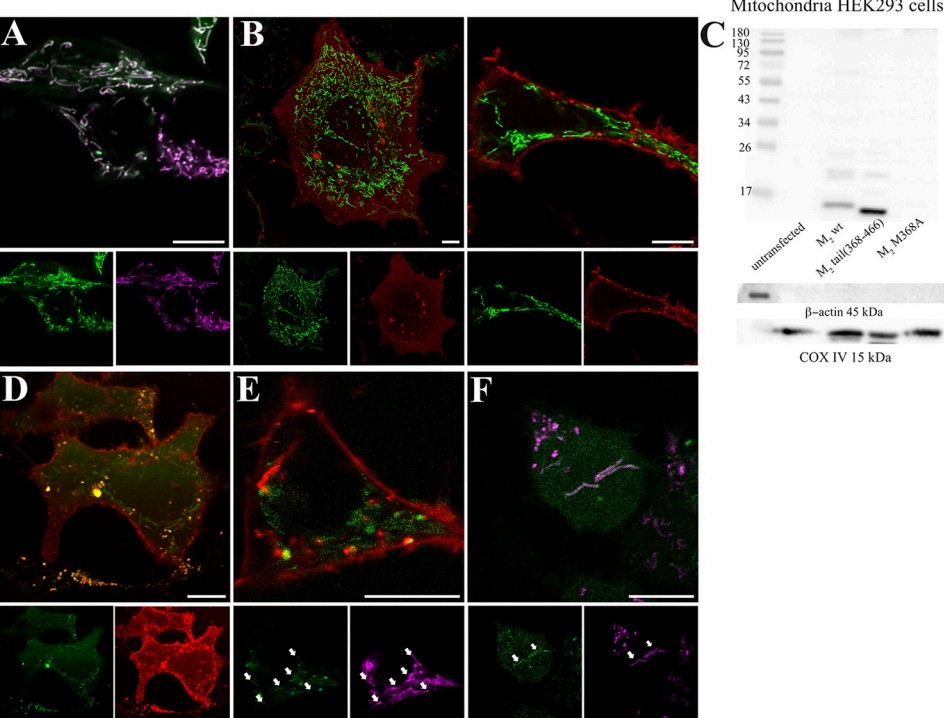

**Fig 3. $M_2$tail(368–466) predominantly localizes to the cell mitochondria.** (**A**) Colocalization of $M_2$tail(368–466)-EGFP (green) together with the mitochondrial targeted dye Mitotracker (magenta) in HEK293 cells. Separate spectral channels are displayed below. (**B**) Co-expression of $M_2$tail(368–466)-EGFP (green) and wild-type $M_2$-mRuby2 (red) in HEK293 cells. Separate spectral channels are displayed below. Left: confocal micrograph of the basal membrane of 2 cells. Right: confocal cross section of a separate cell. (**C**) Western blot (10% TRIS-Glycine PAA gel) of mitochondria purified from HEK293 cells transfected with the $M_2$-myc, $M_2$tail(368–466)-myc, $M_2$M368A-myc, together with an untransfected control are blotted using an anti-myc antibody. Loading control was verified by immunoblotting COXIV and β-actin. (**D**) Cellular localization of the construct $M_2$-mRuby2-STOP-$M_2$-i3-tail-EGFP in HEK293 cells. Expression of the $M_2$tail(368–466)-EGFP (green) after the STOP codon is an indicator of the endogenous IRES activity. While $M_2$-mRuby2 localizes correctly and predominantly to the cell plasma membrane, $M_2$tail(368–466)-EGFP localizes intracellularly. Separate spectral channels are displayed below. (**E**) Same conditions as in D, for a separate cell where Mitotracker (magenta) was added. (**F**) $M_2$-fr.sh.-EGFP (green) expressed in conjunction with Mitotracker (magenta). The EGFP is expressed despite the creation of an in-frame stop after M368, confirming endogenous cap-independent translation of the $M_2$tail(368–466). Arrows indicate areas of colocalization with the mitochondria. Manders Colocalization Coefficient MCC2 is 0.3 ± 0.1 ($n$ = 6). Separate spectral channels are displayed below. Scale bars are 10 μm throughout. IRES, internal ribosome entry site.

$M_2$ muscarinic receptor C-terminally labeled with the red fluorescent protein mRuby2. As expected, and in agreement with what is observed for several other GPCRs [13], the $M_2$ muscarinic receptor expression is homogeneous at the cell membrane, as opposed to the localized EGFP signal at the subplasmalemmal and intracellular mitochondria.

In line with the radioligand binding data shown in **Table 1**, very little expression of $M_2$tail (368–466)-EGFP was observed on the plasma membrane, as we confirmed by fluorescence correlation spectroscopy (FCS) [14] (**S6B–S6D Fig**). In contrast, when the $M_2$tail(368–466)-EGFP was co-expressed with $M_2$stop228, an increased localization could be seen on the cell membrane, suggesting a chaperone activity of the $M_2$ N-terminal domain ($M_2$trunk) (**S6E–S6G Fig**) and providing an explanation for the data obtained in radioligand binding studies and reported in **Table 1**. Likewise, constructs such as $M_2$tail(368–466)-mRuby2 (a red monomeric fluorescent protein) and $M_2$tail(368–466)-tdTomato (a tandem-dimer red fluorescent protein) also displayed clear mitochondrial localization (**S6H and S6I Fig**), ruling out bias due to the fluorophore used. We obtained comparable findings also in COS-7 cells (**S6J Fig**).

Mitochondrial import of the C-terminal fragment of the $M_2$ receptor was further verified by immunoblotting of protein extracts from purified mitochondria of HEK293 cells transfected with myc-tagged $M_2$ receptor, $M_2$tail(368–466), and $M_2$M368A, a mutant where the key methionine for the cap-independent translation of the C-terminus is changed to an alanine. **Fig 3C** shows that in purified mitochondria (unlike in the cytosolic and microsomal supernatants originating from the purification process, **S6K Fig**), only the protein bands associated to the $M_2$ C-terminal fragment were visible. Notably, no mitochondrial import was visible for the $M_2$M368A, confirming that the third in-frame methionine (M368) in the i3 loop of the $M_2$ receptor is necessary for the efficient production of the C-terminal fragment and its localization to the mitochondria. It thus appears that the expression of the $M_2$tail(368–466) fragment in the $M_2$M368A mutant is significantly impaired, as demonstrated also in experiments such as binding assays with the mutant $M_2$stop228/stop368, where binding was completely abolished (**S1 Table**). The slight offset between the lower bands in the $M_2$wt and $M_2$tail(368–466) lanes matches exactly the different length of the linker used to fuse the myc-tag in the 2 constructs (residues TRTRPL for the full-length receptor and only L for the tail construct, a difference of 5 residues leading to an expected MW difference of 0.63 kDa).

## Localization of the $M_2$ C-terminal fragment generated by cap-independent expression under the control of the i3 loop IRES

We then sought to express the $M_2$tail under the control of the i3 loop IRES rather than a viral promoter, to further exclude the possibility that the observed mitochondrial localization of the $M_2$ carboxyl terminal fragment were due to overexpression of the construct. HEK293 cells were transfected with the construct $M_2$-mRuby2-STOP-$M_2$-i3-tail-EGFP which contains the full wild-type $M_2$ receptor, labeled at the C-terminal with mRuby2, additionally fused to the C-terminal part of the $M_2$ receptor starting at nucleotide 685 of the i3 loop, following the stop codon of mRuby2, and ending with a C-terminal EGFP sequence (**S3 Fig**). Through this experimental strategy, the expression of $M_2$tail(368–466)-EGFP is regulated by the IRES, and C-terminal fragment production is therefore close to physiological expression. The expression of EGFP confirmed that the in vivo production of the $M_2$tail(368–466) is regulated by the IRES within the i3 loop. In addition, the processed receptor fragment was eventually observed to be predominantly confined to the cytosol and the mitochondrial network (**Fig 3D and 3E**).

Notably, when the $M_2$-EGFP fusion protein was expressed and the cells were co-stained with Mitotracker, a faint EGFP signal was also observed in the mitochondrial regions (**S7A Fig**), although the expression of the full-length receptor masks this effect and makes

quantification difficult. On the other hand, **Fig 3F** shows that, when the $M_2$fr.sh-EGFP mutant was expressed in HEK293 cells, the IRES-generated C-terminal fragment of the $M_2$ receptor localized to the mitochondria, albeit displaying also a cytosolic component reflecting the fact that the fragment is generated in the endoplasmic reticulum and needs to traffic to the mitochondria. As mentioned above, in this construct the masking effect of the EGFP fused to the full-length receptor is eliminated (see also **S3 Fig**). Similar mitochondrial localization was also observed when we transfected HEK293 cells with $M_2$fr.sh-mRuby2 (**S7B Fig**), ruling out possible effects due to the specific fluorophore used as a tag. Again, comparable localization patterns were observed when the constructs were expressed in a different cell type, such as the H9c2 rat cardiomyoblast cell lines (**S7C Fig**), where Manders colocalization coefficient reflecting EGFP signal from within the mitochondria [15] return a value $MCC2 = 0.73$ for $M_2$tail(368–466), 0.61 for $M_2$fr.sh-EGFP, and 0.93 from $M_2$stop228-EGFP, indicating in all 3 cases a good degree of mitochondrial localization.

## Mechanism of $M_2$tail import into the mitochondria and localization within the organelle

To gain further insight into the dynamics and mechanisms of the $M_2$ C-terminal fragment import within the mitochondria, we first monitored the kinetics of colocalization between mitochondria and EGFP in cells transfected with $M_2$tail(368–466)-EGFP. In cells observed for several hours, starting at 5 h after transfection (**Fig 4A**), the construct was found initially expressed in the perinuclear region, likely the endoplasmic reticulum, indicating that at least the initial step of subcellular trafficking of the $M_2$tail(368–466)-EGFP is the same as for the full-length receptor [16]. At this time (5 h posttransfection), almost no colocalization with the Mitotracker-stained mitochondria was visible. However, at 6 h posttransfection, the colocalization level increased significantly, and $M_2$tail(368–466)-EGFP was either found in the cytosol or in the mitochondria. At 9 h posttransfection, the colocalization at the mitochondria was well visible and we could observe that almost half of the mitochondrial import occurred within 6 h posttransfection (**Fig 4B**).

This value is qualitatively in agreement with what we observed using an in vitro assay to determine the import of the $M_2$ carboxyl terminal into isolated mitochondria. To this end, we used mitochondria form *S. cerevisiae*, which is a robust and well-studied system for protein import assays. **Fig 4C** shows the autoradiography experiment of isolated, intact, mitochondria incubated with $^{35}$S-labeled full-length muscarinic $M_2$ receptor or $M_2$tail(368–466), made using a reticulocyte in vitro translation system. The data show that the full-length receptor was not imported into mitochondria. On the other hand, the $M_2$tail(368–466) displays a clear mitochondrial import already after 5 min and increasing over time, up to 30 min. Interestingly, this import appears to be independent upon the transporters of the outer membrane Tom5, Tom6, Tom7, and Tom70 (**S7D Fig**), whereas the import of an inner membrane protein, the ADP/ATP carrier (AAC), used here as a control, depends on Tom70. Upon extraction with carbonate most of $M_2$tail(368–466) was found in the pellet, and therefore presumably integrated in mitochondrial membranes. Furthermore, Blue native analysis suggests that the mitochondrially imported $M_2$tail(368–466) is not a monomer but rather it is part of a larger complex of approximately 180 kDa (**S7D Fig**). All together, these data indicate that $M_2$tail(368–466) localizes in one of the mitochondrial membranes, to form a complex with other native mitochondrial proteins.

To further discriminate if the construct is on the outer or inner mitochondrial membrane, we set up a fluorescence assay based upon GFP fluorescence complementation [17]. HEK293 cells were first transfected with 2 constructs: $M_2$tail(368–466)-GFP11, in which a short 16

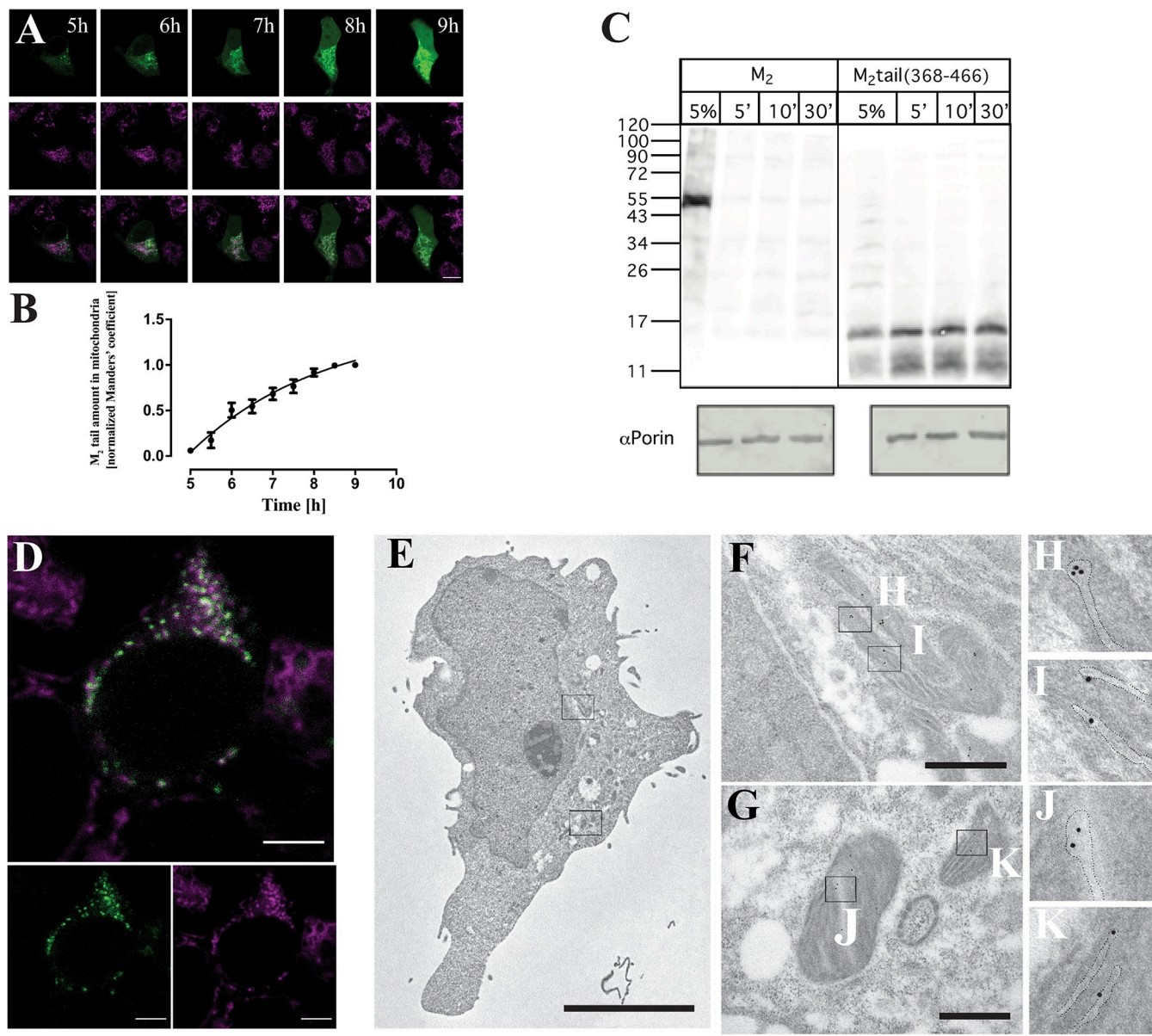

**Fig 4. Mitochondrial import and organelle localization of M₂tail.** (**A**) Time course of M₂tail(368–466)-EGFP import in the mitochondria of a HEK293 cell over a 4 h interval, starting 5 h after cell transfection, as indicated. Panels, from top to bottom show M₂tail(368–466)-EGFP (green), Mitotracker (magenta) and overlay displaying increasing colocalization over time. (**B**) Quantification of the colocalization of the M₂tail(368–466)-EGFP in the mitochondria by calculating the Manders Coefficient MCC2 ($n = 7$ cells, from 3 independent transfections, error bars are SEM). Values were normalized to the maximal MCC2 value. (**C**) Import of protein M₂tail(368–466) in isolated yeast mitochondria: $^{35}$S-labeled and translated in vitro full-length muscarinic M₂ receptor or M₂tail(368–466) expressing 5 C-terminal methionines were with isolated yeast mitochondria for 5, 10, and 30 min in the presence of the inner membrane potential. The 5% lane represents a control with the pure protein, with no mitochondria. αPorin is used as a loading control. (**D**) Detail of a HEK293 cell expressing the M₂-GFP11 construct together with mito-GFP1-10. The M₂-GFP11 was sequentially transfected 24 h after the mito-GFP1-10 Magenta indicates Mitotracker. White signal in the overlay indicates M₂tail(368–466) trafficking to the organelle. All confocal micrographs were obtained by sequential imaging using 488 nm excitation and detection in the 520–600 nm (EGFP), and at 633 nm and detected in the 650–700 nm range (Mitotracker) using HyD detectors in Photon Counting Mode. Scale bars are 10 μm throughout. (**E**) Representative IEM micrograph of a COS-7 cell showing a large nucleus, several mitochondria, and vacuoles nearby. Bar: 5 μm. (**F, G**) High magnification of portion of the cell cytoplasm showing a few gold particles confined to the mitochondria. Bar: 500 nm. (**H–K**) Gold particles appear associated with the mitochondrial cristae, highlighted by the dotted line. A total number of 236 gold particles in 40 mitochondria from 19 sections were detected. Controls reported in S7E Fig. Source data for panels B, and raw images related to panels A and D can be found in S1 Data. IEM, immunoelectron microscopy.

amino acids peptide of GFP was fused at the C-terminus of $M_2$tail(368–466), and mito-GFP1-10, in which the Cytochrome C Oxidase Subunit 8A (COX8A) was N-terminally fused to the other GFP fragment containing the remaining 10 beta strands (see **Materials and methods**). **S8A Fig** shows the successful recombination of the 2 proteins within the mitochondrial matrix. This observation is supported by a set of controls that include a mitochondrial intermembrane space-targeted protein achieved using the targeting sequence of SMAC [18]. **S8B–S8D Fig** show a set of positive recombination controls using respectively SMAC-GFP11+SMAC-GFP1-10, mito-GFP11+mito-GFP1-10, and $M_2$tail(368–466)-GFP11+$M_2$tail(368–466)-GFP1-10, all displaying mitochondrial recombination comparable to what is observed in **S8A Fig** for $M_2$tail(368–466)-GFP11+mito-GFP1-10. As negative controls, we observed the lack of complementation of $M_2$tail-GFP11 with SMAC-GFP1-10 (**S8E Fig**), and the absence of any significant signal when mito-GFP1-10 was expressed alone (**S8F Fig**).

We then applied this approach to the full-length $M_2$ receptor, by generating the $M_2$-GFP11 construct and by transfecting it in HEK293 cells. **Fig 4D** shows a green signal from the recombined GFP visible in the mitochondria. All together, these data suggest that the $M_2$ C-terminal fragment is localized in the inner mitochondrial membrane, with its own C-terminal domain facing the matrix. This interpretation is also finally confirmed by immunoelectron microscopy (IEM) imaging of thin sections from cells expressing the $M_2$stop228 C-terminally labeled using an myc-tag after immune-gold labeling (**Fig 4E**). As shown in **Fig 4F and 4G**, gold particles were observed only in transfected cells (**S7E Fig**), and exclusively in mitochondria, and at this stage they appeared localized either to the mitochondrial matrix or to the lumen of the cristae (**Fig 4H–4K**). The count of positive mitochondria per cell and the average number of particles per mitochondrion are notably low. However, as we will discuss later, this indicates a stringent control over the expression of the $M_2$tail fragment in these organelles. As we will elaborate further, the $M_2$tail fragment exhibits a profoundly detrimental impact on cell viability and mitochondrial respiration.

## Cell stress up-regulates IRES-mediated $M_2$tail expression and drives mitochondrial localization

Having established that $M_2$tail localizes in the inner mitochondrial membrane, we then moved to look at the effect of stress on the level of expression of the C-terminal fragment. Stressed cells are known to attenuate their translational activity via phosphorylation of the initiation factor eIF2α. However, certain mRNAs may be insensitive to eIF2α phosphorylation level if their translation is regulated by IRES, and under these conditions their expression is bound to increase rather than to attenuate [19,20].

Therefore, we transfected COS-7 cells with the Sirius-$M_2$i3(417n)-EGFP reporter construct and stressed by starvation for 3 h by replacing the culture medium with HBSS or phosphate-buffered saline (PBS). Notably, PBS has a poorer nutrient content than the HBSS media, thus causing a more pronounced starvation. During this time interval, the green fluorescence was seen to gradually increase up to 1.3 times the basal level observed in HBSS-starved cells and 1.5 times for PBS-starved cells (**Fig 5A**), confirming that the C-terminal fragment expression is up-regulated in response to stress conditions such as cell starvation. The same experiment was reproduced in single cells, displaying an increase of the average EGFP/Sirius ratio (slope), and thus of IRES-mediated translation, upon HBSS starvation in HEK293 cells (**Fig 5B**). We further used the GFP fluorescence complementation assay to emphasize the importance of stress in the pattern of expression of the $M_2$ gene. HEK293 cells were co-transfected with $M_2$-GFP11 and Mito-GFP1-10. HEK293 cells grown in culture medium predominantly expressed the full-length $M_2$ receptor protein (**Fig 5C**).

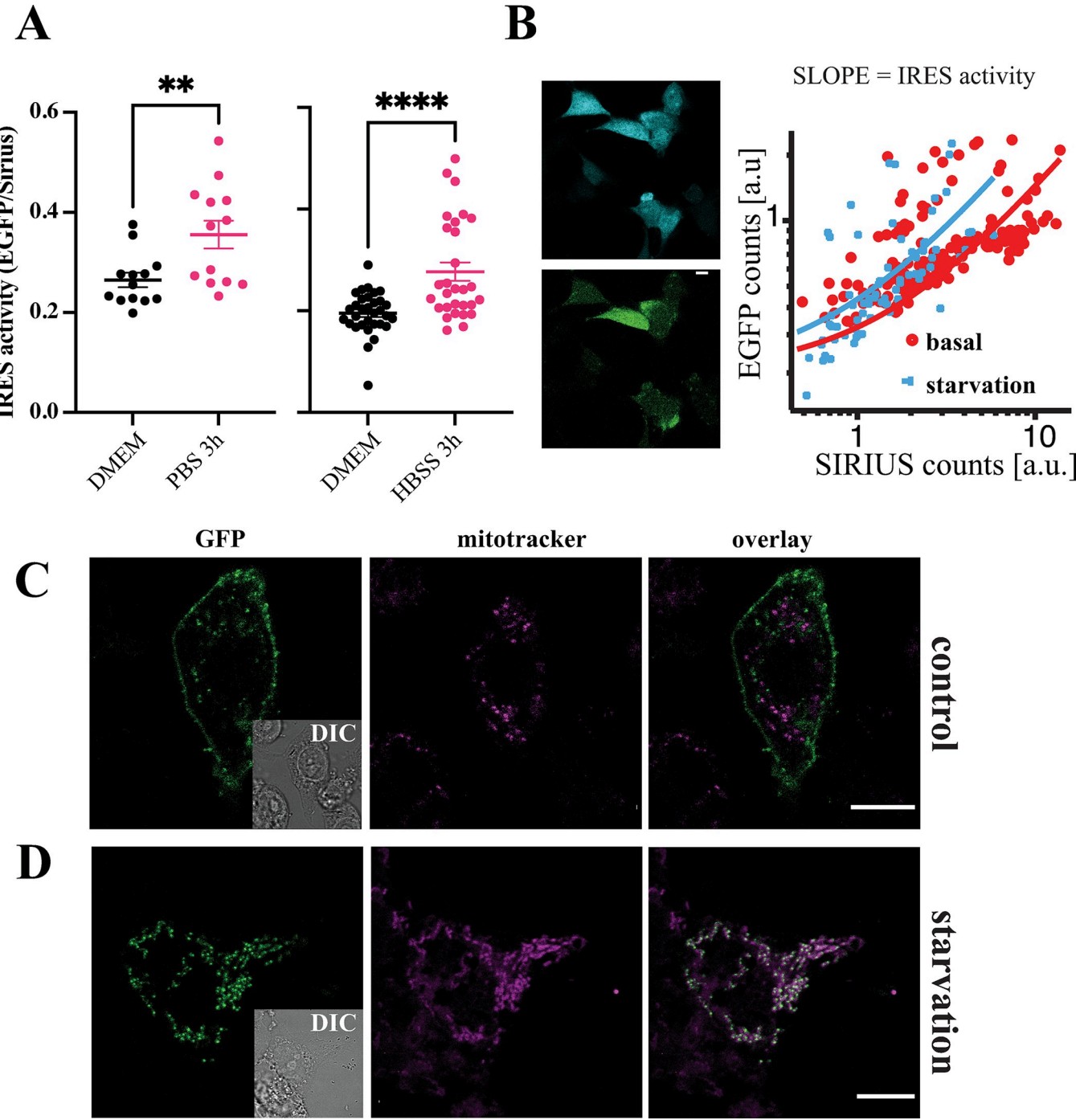

**Fig 5. Effect of integrated stress response on IRES activity and M₂tail(368–466) localization.** (**A**) 96-well plate reader measurement of the increase of EGFP fluorescence relative to Sirius in COS-7 cells transfected with the bicistronic Sirius-M₂i3(417n)-EGFP plasmid and starved for 3 h respectively in PBS (left panel) ($n = 3$ transfections, 16 wells) and HBSS (right panel) ($n = 1$ transfection, 16 wells). Significance values reported in the graphs were determined by a one-tailed Student $t$ test, $p$-values: **** $p < 0.0001$; ** $0.001 < p < 0.01$. (**B**) Single-cell measurement of EGFP vs. SIRIUS expression in HEK293 cells, in basal conditions and upon starvation. Lines are linear fits (note log-log scale). (**C**) HEK293 cells were sequentially transfected with pMito-GFP1-10 (24 h) and M₂-GFP11 (8 h, of which 6 h under starvation conditions) and then grown in full culture medium or (**D**) starved for 6 h after transfection of M₂-GFP11. Shown are EGFP fluorescence (left, with DIC inset), Mitotracker (center), and overlay (right). Scale bars are 10 μm. Source data for panels A and B, and raw images related to panel C and D can be found in S1 Data. IRES, internal ribosome entry site; PBS, phosphate-buffered saline.

The green fluorescence detected at the plasma membrane is explained with the chaperoning effect of $M_2$-GFP11 on Mito-GFP1-10 before its sorting to the mitochondria. In sharp contrast, when HEK293 cells were first transfected with Mito-GFP1-10 and, 24 h later, transfected with $M_2$-GFP11 followed by a 6 h starvation in HBSS, the pattern of expression changed drastically: IRES-mediated translation of the $M_2$tail(368–466) fragment led to a predominantly mitochondrial green fluorescence signal of the complemented GFP (**Fig 5D**). These results are in line with our observation that the IRES-mediated C-terminal fragment localization to the mitochondria in cells expressing the construct $M_2$-mRuby2-STOP-$M_2$-i3-tail-EGFP was enhanced upon cell starvation, and drastically reduced upon mutation of the M368 (**S9 Fig**).

## Functional role of the C-terminal fragment in the mitochondria

The next step of our investigation concerned the obvious question about the functional effects that the observed localization of the $M_2$ C-terminal fragment might have on the mitochondrial activity, and—more generally—on the energy metabolism of the cell. To investigate this aspect, we measured the extent of oxygen consumption in COS-7 cells expressing $M_2$-EGFP, $M_2$tail (368–466)-EGFP, or the sole EGFP as a control. Initially, we normalized the expression of the 3 proteins to the amount of EGFP expressed in the cells, then the metabolic functions were assayed on a Seahorse instrument using cells transfected with normalized amounts of plasmids (**Fig 6A**).

Our data did not show a significant difference in total oxygen consumption capacity between cells transfected with cytosolic EGFP and the untransfected control. However, a significant decrease of oxygen consumption was found in cells transfected with $M_2$-EGFP and even more so with $M_2$tail(368–466)-EGFP (**Fig 6A**). Notably, the basal oxygen consumption rate (OCR) in starvation-induced stress conditions was profoundly decreased in $M_2$tail(368–466) cells. This occurred to a smaller extent also in $M_2$ cells compared to EGFP-transfected and empty control cells (see also the time-course in **S10A Fig**). These results are in line with what was observed using a Clarke-type oxygen electrode (**S10B and S10C Fig**).

Notably, there was no difference in the number of apoptotic cells among the different cell groups as shown by the caspase-3 activation assays of apoptosis [21] (**S10D Fig**) and morphological investigation of the mitochondrial network (**S10E Fig**). To test whether the $M_2$tail effects on mitochondrial oxygen consumption could alter the production of ROS, the measurements were combined in a single-cell assay. Remarkably, ROS production (assayed using a fluorescence readout kit, where the fluorochrome becomes activated in contact with oxidized species) was clearly reduced within and in proximity of mitochondria of cells expressing a high level of the $M_2$tail(368–466) (**Fig 6B** (**inset**)). By regrouping individual mitochondria in classes according to the level of the ROS marker expression, an inverse correlation between the amount of receptor fragment (Green Fluorescence) and the production of ROS (Red Fluorescence) was revealed (**Fig 6B**).

Cell growth was also found to be affected depending on which of the constructs was transfected. Notably, transfection of HEK293 cells with full-length $M_2$ receptor reduced significantly cell growth compared to the control, whereas cells transfected with the mutant $M_2$M368A, that do not express the $M_2$ C-terminal fragment (**Fig 6C**), displayed a significantly higher growth. All cells were incubated with saturating concentration of the $M_2$ antagonist atropine throughout, in order to exclude effects on cell growth and proliferation due to the canonical receptor signaling.

These functional data are further supported by Fluorescence Resonance Energy Transfer (FRET) data aimed at investigating the molecular interactions of the mTurquoise2-labeled $M_2$tail(368–466) with mitochondrial proteins. Based on the oxygen consumption data

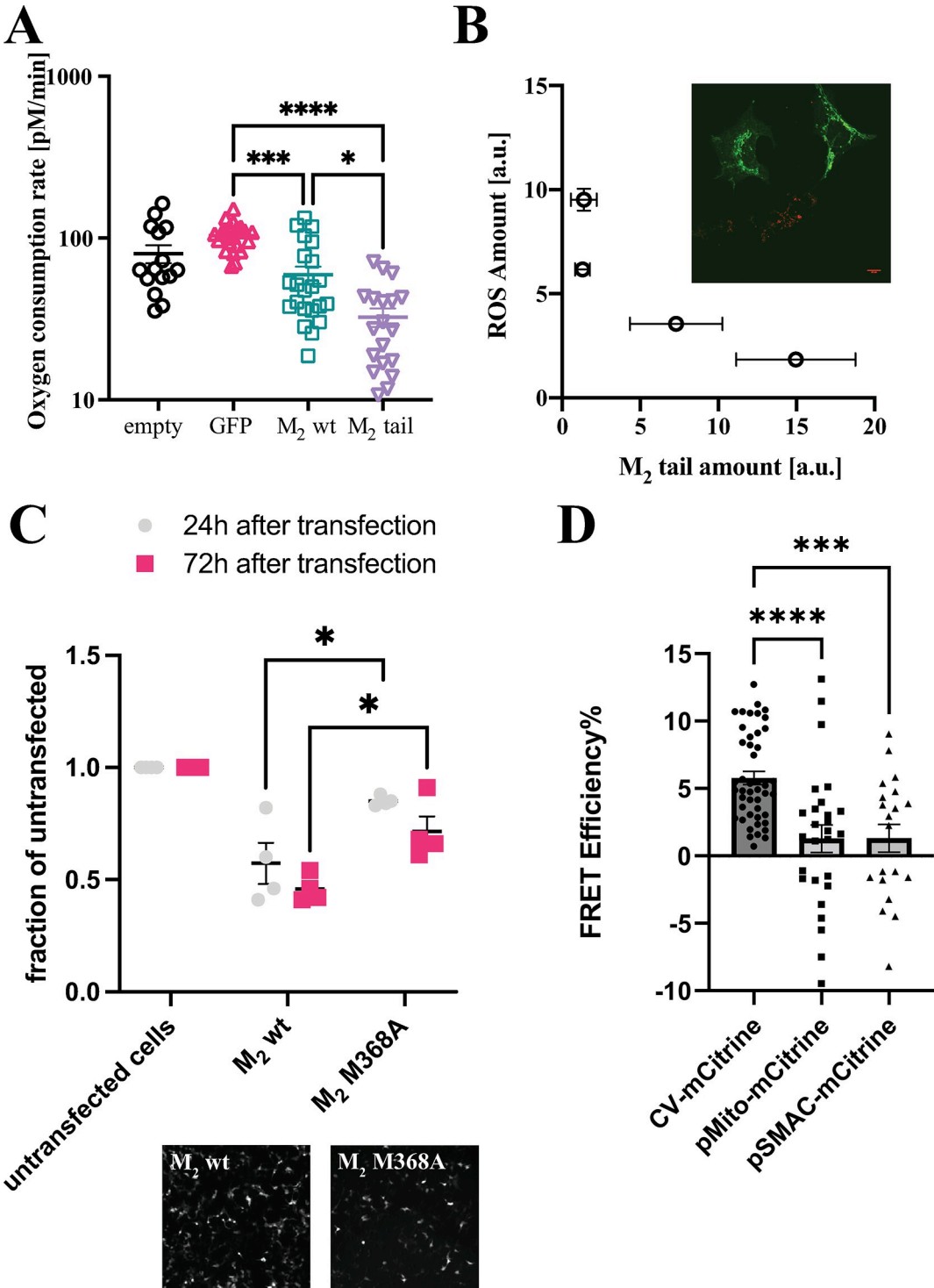

**Fig 6. Effects of M₂tail expression, or absence thereof, on mitochondrial function and cell metabolism in transfected HEK293 cells.** (**A**) Basal oxygen consumption upon HEK293 cell stress—following starvation—measured by a Seahorse assay at a constant temperature of 37°C. Significance values reported in the graphs were determined by a one-tailed Student *t* test, *p*-values: **** $p < 0.0001$; *** $0.0001 < p < 0.001$; * $0.01 < p < 0.05$. (**B**) Localization of M₂tail(368–466)-EGFP in HEK293 (green) cells stained with a marker for ROS (Cell Rox Deep Red). Relationship between green (M₂tail abundance) vs. red (amount of ROS produced) fluorescence intensities measured from regions of the mitochondrial network. The data were binned in 4 groups as a function of the amplitude of Cell Rox Deep Red signal (0–2.5, 2.5–5, 5–7.5, and 7.5–10 intensity counts). (**C**) Cell viability assay performed on HEK293 cells transfected with M₂ and M₂M368A, compared to an untransfected control. Cells

were counted 24 h and 72 h after transfection in T-25 flasks, exposed throughout to 100 nM atropine. The experiments were performed in 2 biological replicates, each with 2 technical replicates. Transfection efficiency (to the right of the chart) was monitored thanks to the mRuby2 fluorescent reporter within a bicistronic cassette within each vector (see **Materials and methods**). Significance values reported in the graphs were determined by a one-tailed Student $t$ test, $p$-values: $**$ $0.001 < p < 0.01$; $*$ $0.01 < p < 0.05$. (**D**) FRET efficiency plot of the interaction between $M_2$tail(368–466)-mTurquoise2 and $F_0F_1$ATP Synthetase (CV-mCitrine), COX8a(pMito)-mCitrine, and SMAC-mCitrine, as determined by acceptor photobleaching measurement in a confocal microscope. Each dot represents a ROI within a cell. At least 3 cells from 2 separate transfections were imaged for each pair. Source data for panels A–D can be found in S1 Data. ROS, reactive oxygen species.

(**Figs 6A** and **S10A**), we chose as primary target the $F_0F_1$ ATP Synthetase complex, also indicated as complex V (CV), and we labeled the γ subunit with mCitrine, a fluorescent acceptor. The γ subunit is located towards the mitochondrial matrix. We further monitored interactions with an intermembrane space protein (SMAC) and a protein localized to the mitochondrial matrix (COX8A), both fused to the mCitrine fluorescent acceptor. As clearly illustrated in **Fig 6D**, FRET efficiency, an indicator of molecular interaction, is significantly larger for $M_2$tail (368–466)-CV pair than both the intermembrane space SMAC and the matrix COX8A. The latter result is particularly relevant, as it indicates a specific interaction of $M_2$tail(368–466) with CV.

## Relevance of C-terminal expression and mitochondrial translocation in a physiological, endogenous setting

The results provided in the previous sections convincingly demonstrate the cap-independent translation of the $M_2$ C-terminal fragment, its up-regulation and mitochondrial localization upon stress response, and the subsequent effect in regulating oxidative phosphorylation by a direct interaction with complex V of the respiratory chain. To test the physiological relevance of these findings, we first prepared protein extracts from heart, brain, and spleen tissues obtained from a line of knock-in mice (see **Materials and methods** and **S1 Methods**), designed to have expression of a Chrm2/tdtomato fusion protein driven by the endogenous $M_2$ muscarinic acetylcholine receptor promoter/enhancer sequences. The heart and brain are known to express sizable levels of the $M_2$ receptor, whereas the spleen, where no expression of the $M_2$ receptor is reported in the proteomic databases, was used as a negative control.

The use of a transgene tag was necessary, since antibody against the $M_2$ receptor are notoriously ineffective when targeted at the C-terminal portion of the receptor. Instead, the tdtomato fluorescent protein can be targeted by a specific polyclonal antibodies as shown by the western blot displayed in **Fig 7A**. As expected, the full-length receptor-tdtomato fusion was observed at approximately 130 kDa (approximately 75 kDa contributed by the glycosylated receptor and approximately 55 kDa contributed by tdtomato). Notably, we observed again (both in the heart and brain, but not in the spleen) a specific band doublet, this time at approximately 75 kDa, which corresponds to the sum of the molecular weights of the tail fragment added to the molecular weight of tdtomato. When mitochondria were purified from the heart and spleen of the Chrm2/tdtomato mice (**Fig 7B**), the band at approximately 75 kDa was substantially enriched, matching the expected molecular weight of the $M_2$tail. These data suggest that the $M_2$tail is expressed endogenously, and it is sorted to the mitochondria.

We finally posed the question of what would happen abolishing the translocation of the $M_2$ C-terminal fragment to the mitochondria, in a physiological and entirely label-free setting. Notably, hiPSCs combined with genome editing approach are state-of-the-art tools to investigate the effect of gene mutations in physiological and label-free setting. Therefore, we incorporated by Crispr/Cas9 the homozygous M368A mutation in hiPSCs (**Materials and methods**).

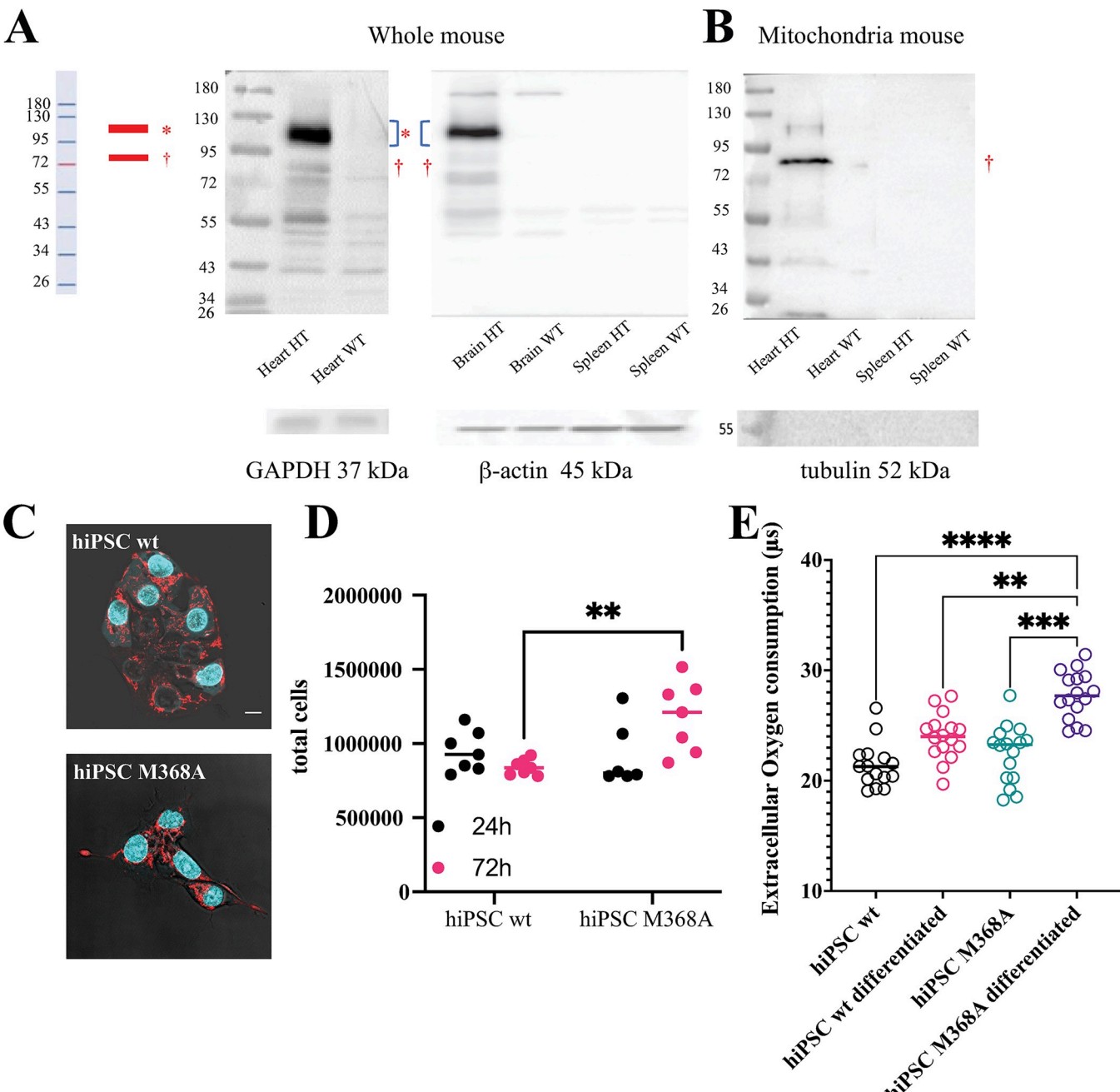

**Fig 7. IRES-mediated M₂tail(368–466) expression occurs in endogenous, physiological settings and its suppression affects cell metabolism.** (**A**) Western blot from tissue preparations originating from the indicated organs of M₂-tdtomato knock-in mice (HT) and wild-type animals (WT). Immunodetection was performed using an anti-tdtomato primary antibody. Loading control normalization was performed using GAPDH in the heart, and β-actin in the brain and spleen. (**B**) Western blot (8% TRIS-Glycine PAA gel) of mitochondria purified from tissue preparations originating from the indicated organs of M₂-tdtomato knock-in mice (HT) and wild-type animals (WT), detected using an anti-tdtomato antibody. Mitochondrial preparation purity was assessed by blotting tubulin. (**C**) Confocal micrographs of wt hiPSCs (left) and Crispr/Cas9 clone harboring the M368A mutation within the M₂ gene (right), stained with DAPI (cyan) and Mitotracker (red). (**D**) Cell proliferation assay performed for wt hiPSCs and M368A hiPSCs grown in T25 flasks, exposed to atropine. Cells were counted 24 h and 72 h after seeding. Significance values reported in the graphs were determined by a one-tailed Student $t$ test, $p$-values: ** $0.001 < p < 0.01$. (**E**) Extracellular oxygen consumption assay, performed on wt and M368A hiPSCs, prior to and after 7 days undirected differentiation under atropine exposure. The lifetime of a commercial dye increases as oxygen in the medium is progressively consumed by the cells. The data displayed was collected after 60 min incubation. Significance values reported in the graphs were determined by a one-tailed Student $t$ test, $p$-values: *** $0.0001 < p < 0.001$; ** $0.001 < p < 0.01$. Source data for panels D and E can be found in S1 Data. hiPSC, human-induced pluripotent stem cell; IRES, internal ribosome entry site.

Based on our data of **Fig 3C**, this mutation should be sufficient to significantly impair the M$_2$tail production and mitochondrial translocation. Wild type and M368A hiPSCs were cultured in pluripotency maintaining medium, and subsequently undirectedly differentiated upon exposure for 7 days to DMEM-F12 medium, as described in the **Materials and methods**. **Fig 7C** shows confocal micrographs for wild-type and M368A hiPSCs cultured in E8 medium, respectively, with the M368A hiPSCs displaying a mildly more spread-out phenotype. We shall note here that the M$_2$ receptor has been found to be expressed in hiPSCs lines, and it has been shown to play a role in stem cell differentiation [22,23]. In this context, however, we were not interested in the canonical signaling of the receptor, and all experiments were performed with a saturating concentration of atropine, to block any possible effect of the canonical signaling of the M$_2$ receptor. We then evaluated cell proliferation, observing a significant increase in the proliferation of the M368A clone 72 h after seeding **Fig 7D**. The observation that cells harboring the mutation that suppresses M$_2$tail display an increased proliferation suggests that the same effect is at play here, as observed in the overexpression system (**Fig 6C**). These results are supported by extracellular oxygen consumption assays performed, for both hiPSCs and differentiated hiPSCs (**Fig 7E**): the M368A cells indeed display an accelerated OCR, and this effect increases as the cells progressively differentiate.

For comparison, the assay was also carried on HEK293 overexpressing the wild-type M$_2$ receptor and the mutant M$_2$M368A (**S10F Fig**). We shall further note here that signaling properties of M$_2$ M368A with respect to the wild-type M$_2$ receptor are comparable, as displayed by pERK immunoblotting as well as FRET G$_i$ activation assays performed in cells overexpressing either the M$_2$wt or the M$_2$ M368A (**S10G and S10H Fig**).

## Discussion

Over the last 10 years, growing evidence has accumulated indicating that GPCRs can be functional and signal also at cellular locations other than the plasma membrane. These locations can be associated to their canonical trafficking pathways (such as sorting vesicles and endosomes) [24], but also to non-canonical locations, such as the nucleus, the nuclear membrane, the Golgi apparatus, and the mitochondria (reviewed in Jong and colleagues [25]). Some of these locations are surprising, since many ligands are non-membrane permeable, and it is still debated how they can reach their target receptors at such intracellular locations.

Interestingly, the specific localization of these receptors appears to go hand in hand with a specific function. In these cases, activation/inhibition of mitochondrially localized GPCRs have been shown to specifically affect the organelle function, ranging from OCRs to Ca$^{2+}$ handling and even to apoptosis. Notably, Bénárd and colleagues [7] showed that the CB$_1$R on the outer mitochondrial membranes affects neuronal energy metabolism; Suofu and colleagues [6] demonstrated that activation of the MT$_1$R by a mitochondrial melatonin pool can affect cytochrome C release. This is particularly interesting, since cytosolic cAMP does not appear to be able to activate mitochondrial PKA, and hence, a mitochondrially localized GPCR signalosome appears to be required to enable such downstream signaling cascade [26].

However, other non-canonical mechanisms are also possible for the regulation of mitochondrial function. It is well known that GPCRs are subject to interindividual variability, arising, for example, from mRNA alternative splicing (i.e., the process which allows the recombination of exons) that leads to different isoforms which may present distinct signaling and localization properties [27]. Receptor variants with different functional and localization properties may also occur due to other processes. For example, our comprehension of eukaryotic translational regulation is facing a rapid expansion, as exemplified by the observation that

small open reading frames (smORFs), also within long non-coding RNAs (lncRNAs), are an important class of genes, oftentimes with mitochondrial localization [28].

In this study, for the first time, we provide evidence that an IRES sequence drives expression of a membrane receptor fragment and acts as a regulator of its expression levels and function. IRES sequences are distinct regions of the mRNAs that are capable of recruiting ribosomes in the proximity of an internal initiation codon and translate transcripts in a cap-independent manner [28–32]. While in eukaryotes, the most common mechanism of translation initiation is the cap-dependent process, which requires the interaction of several key proteins at the 5′-end (5′ cap) of mRNA molecules, recent years have seen growing evidence that IRES sequences, which are common features of virus replication [32], can mediate cap-independent translation of eukaryotic mRNA(s) [30].

The IRES sequence is located within the third intracellular loop of the $M_2$ muscarinic receptor, where it plays a crucial role in regulating the expression of its carboxy-terminal domain, which we identified as $M_2$tail(368–466).

We assessed the IRES-driven expression of the C-terminal fragment through a broad set of constructs, meant to exclude alternative synthesis mechanisms such as stop-codon read-through, and a diversified palette of techniques. Radioligand binding data (**Table 1**) pointed to the necessity of the C-terminal fragment synthesis, to explain the observed downstream signaling in mutants carrying a stop codon in the initial portion of the i3loop of the $M_2$ receptor, upstream of M368. Fluorescence microscopy allowed us visualizing the production of the EGFP-tagged C-terminal fragment in these receptor mutants (**Fig 1**).

Mutagenesis of key residues within the i3 loop of the $M_2$ receptor allowed us to pinpoint the third in-frame methionine as the most likely candidate for the beginning of the cap-independent translation (**Fig 2**). This observation appears to be supported by the observation of a peak in ribosome coverage around the M368 seen in ribosome profiling databases (as reviewed in Ingolia and collegaues [33]), and particularly prominent in those from left ventricle cardiomyocytes, where $M_2$ receptor expression is known to be abundant [34].

To our surprise, in contrast to the plasma membrane localization of the wild type, full-length $M_2$ receptor, the C-terminal fragment was localized mostly in mitochondria in HEK293 cells. This localization occurred markedly when overexpressing the C-terminal fragment $M_2$tail(368–466), but could be also clearly detected when the C-terminal fragment was expressed under the control of the IRES sequence of the full-length receptor (**Fig 3**). These results were further validated in a rat cardiomyoblast cell line such as H9c2 (**S7C Fig**).

We devised several strategies to detect the localization of the IRES-driven fragment in the mitochondria by avoiding the cell-wide background signal from the full-length receptor. **Fig 3D–3F** show 2 prominent examples, namely relying on spectrally unmixing the 2 components, or by sending out of frame the EGFP at the C-terminus of the full-length receptor. By studying mitochondria purified from HEK293 cells, we obtained an initial confirmation that the mutation of the M368, which we identified as the starting point of the IRES-mediated translation, is sufficient to suppress any mitochondrial targeting of the $M_2$ C-terminal fragment, and most likely, its synthesis altogether (**Fig 3C**). We shall note here that while we cannot rule out the possibility that the lack of detection of this fragment in mitochondria in western blots may be attributed to the overall lower expression of the mutant $M_2$M368A receptor in HEK293 cells our conclusion is supported also by radioligand binding data (**S1 Table**) as well as by gene editing in hiPSCs, as discussed below.

We then addressed the mechanism of mitochondrial import of the C-terminal fragment and its localization within the organelle (**Fig 4**). Our data, originating both from live cell import kinetics as well as from in vitro import into isolated mitochondria (**Fig 4A and 4B**), support the notion that the fragment becomes rapidly imported into mitochondria following

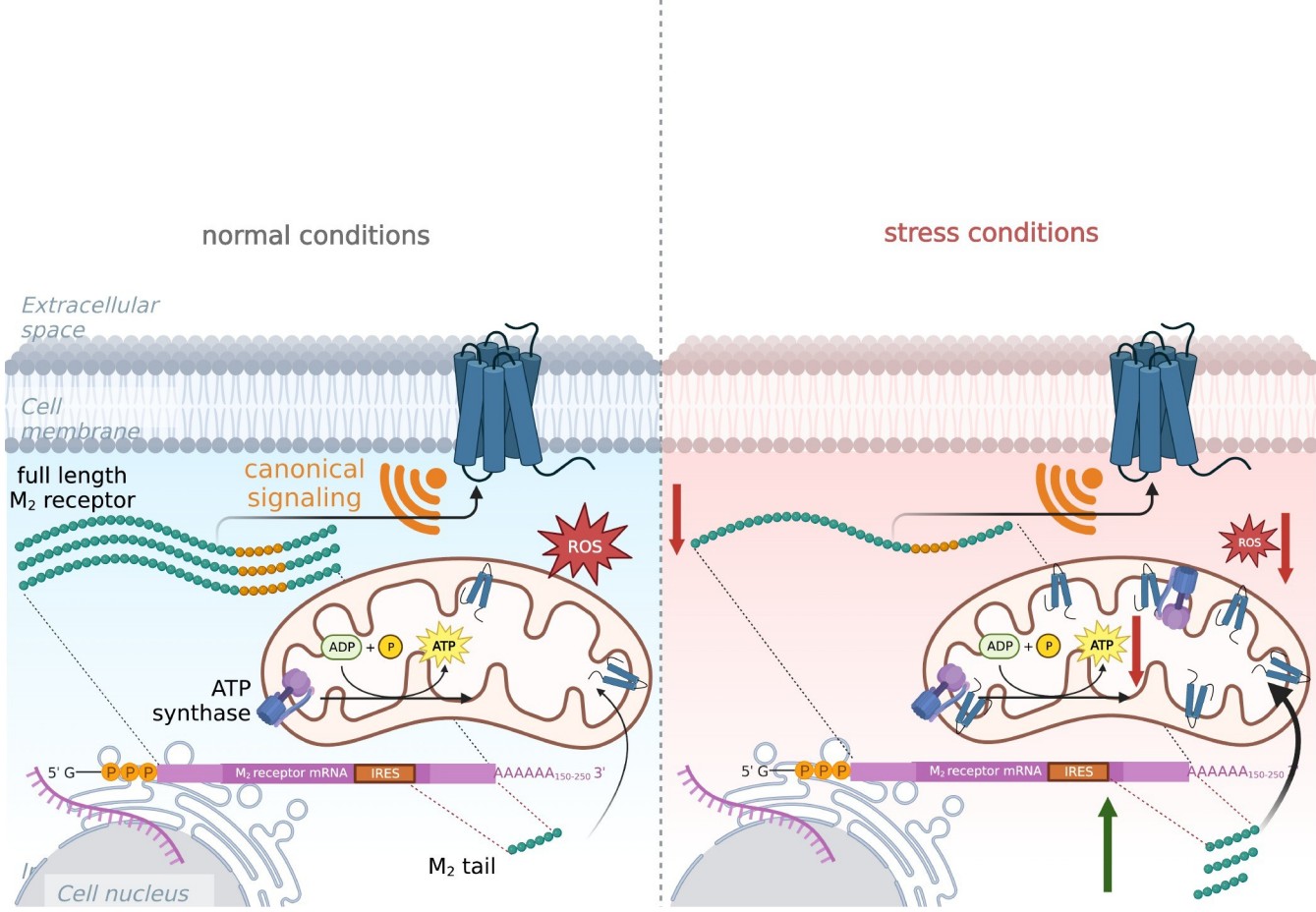

**Fig 8. Schematic representation of the wild type M$_2$ receptor gene products, their destination in the cell and their function.** The M$_2$ receptor gene gives rise to 1 transcript but 2 different translation outputs: when translation starts from the canonical cap site the M$_2$wt receptor is generated (full-length M$_2$ receptor). When translation is driven by the IRES, the C-terminal fragment is generated (M$_2$ tail). The 2 proteins are then sorted in 2 different cell compartments where they play completely different roles: the M$_2$ receptor signals canonically at the plasma membrane as a G protein-coupled receptor. The M$_2$tail is instead sorted to the mitochondria. In normal conditions, the activity of the IRES is low, and only a small amount of M$_2$tail is produced in the cell (left). However, upon cell stress conditions (right), M$_2$tail IRES-mediated translation is up-regulated and M$_2$tail in the mitochondria interacts with the ATP synthase and reduces oxidative phosphorylation and ROS formation. Created with BioRender. IRES, internal ribosome entry site; ROS, reactive oxygen species.

translation in the cytosol. Once inside the mitochondria, it undergoes processing leading to the appearance of a lower molecular weight band, observed both in western blots from whole cell and mice tissue protein extracts, as well as from purified mitochondria and in the yeast in vitro assays (**Fig 4C**). The fragment is found predominantly in a carbonate-insoluble fraction, i.e., a membrane, and it appears to form a high molecular weight complex of approximately 180 kDa with other mitochondrial proteins (**S7D Fig**).

Fluorescence complementation assays (**Fig 4D**) and electron microscopy (**Fig 4E–4H**) suggest that the M$_2$ C-terminal fragment predominantly resides in the inner mitochondrial membrane, with its own C-terminal domain exposed towards the matrix. Future work should clarify the precise import pathway.

Under integrated stress response, namely amino acid starvation, IRES-mediated production of the C-terminal fragment was increased (**Fig 5A and 5B**), with a marked increase in mitochondrial localization (**Fig 5C**). Most importantly, our data show that mitochondrial oxygen

consumption was reduced by the $M_2$tail(368–466) (**Fig 6A and 6B**), and cell proliferation was affected (**Fig 6C**). On these bases, the $M_2$tail might be needed in mitochondria as a means to reduce mitochondrial basal oxygen consumption and ATP-linked respiration. An important consequence of reduced respiration is the lower level of ROS produced in mitochondria, as we observed in agreement with this hypothesis (**Fig 6B**).

This observation was confirmed in physiological conditions by western blots in a knock-in mice carrying a tag at the C-terminus of the endogenous $M_2$ receptor. These could show mitochondrial enrichment of a band corresponding to the molecular weight of the C-terminal domain (**Fig 7A**). Moreover, the physiological relevance of our observations was tested in an entirely label-free assay, by Crispr/Cas9 editing of pluripotent stem cells in order to suppress the synthesis of the $M_2$ C-terminal fragment. By mutating the IRES starting methionine M368, we observed an increase of cell proliferation of this clone compared to its wild type counterpart, together with an increase of OCRs (**Fig 7B–7D**). Overall, this corroborates our hypothesis that the $M_2$ C-terminal fragment reduces oxygen consumption upon reaching the mitochondria. The effect was further amplified when the cells were undirectedly differentiated, and it was consistent with the shift of the cell phenotype towards oxidative phosphorylation [35].

The mechanism by which the $M_2$tail(368–466) may reduce oxygen consumption in mitochondria and its meaning needs to be further explored. Our FRET data (**Fig 6D**) indicate clearly that the $M_2$ C-terminal fragment has a specific interaction with the $F_0F_1$ ATP-synthase complex, supporting the concept that it inhibits the activity of this enzyme, thus leading to the observed reduction in OCRs. Future work should clarify the precise import pathway and the inhibitory effect of $M_2$tail(368–466) on $F_0F_1$ ATP-synthase. Nonetheless, since ROS production is implicated in the pathogenesis of several diseases [35], we would propose the hypothesis that increased localization of the $M_2$ C-terminal fragment in mitochondria, as it occurs under integrated stress response, might have protective effects (**Fig 8**).

In broader terms, protection of mitochondria by unconventional import mechanisms to allow localization within the organelle of specific proteins has been previously observed for cells that undergo stress conditions [36]. The mechanism of IRES-dependent modulation of translation and consequent mitochondrial targeting of the translation product of a GPCR may provide another means, completely unrelated to the canonical signaling function of this class of membrane receptors, of a generic cellular response to overcome different types of stress by modulating the mitochondrial proteome and adjusting the mitochondrial bioenergetic state.

## Materials and methods

### Generation of mutant muscarinic $M_2$ receptors for radioligand-binding studies, western blotting, immunohistochemistry, and fluorescence microscopy

Multiple receptor mutants were constructed starting from the human $M_2$ muscarinic receptor expressed in the pcD vector [37]. The constructs are sketched in **S3 Fig** and their cloning is described in detail in the **S1 Methods**.

### Reverse transcription

Reverse transcription was performed using a 24-nucleotide antisense oligo (TTACCTTGTA GCGCCTATGTTCTT) directed against the last 24 nucleotides at the 3′ end of the $M_2$ open reading frame. cDNA amplification was carried out with the same antisense oligo used for reverse transcription, together with a 24 nucleotides sense oligo

(ATGAATAACTCAACAAACTCCTCT) directed against the first 24 nucleotides at the 5′ end of the $M_2$ open reading frame.

## Cell culture and transfection

COS-7 (African green monkey kidney fibroblast-like SV40 transformed cell line) and CHO (Chinese Hamster Ovary) cells (from American Type Culture Collection) were grown at 5% $CO_2$ and 37°C, in high glucose Dulbecco's Modified Eagle Medium (DMEM) supplemented with 10% fetal bovine serum (FBS), 2 mM L-glutamine, 50 IU/ml penicillin, and 50 mg/ml streptomycin, and non-essential amino acids. HeLa cells (from American Type Culture Collection) were cultured in the same medium, but with the double amount of L-glutamine. SHSY-5Y cells were cultured in Roswell Park Memorial Institute (RPMI) medium supplemented with 10% FBS, 2 mM L-glutamine, 50 IU/ml penicillin, and 50 mg/ml streptomycin. Media and reagents used for cell cultures were from Euroclone, PAN Biotech, Biochrome, and Sigma-Aldrich. Cell transfection was performed using FuGene transfection reagents (Promega) as recommended by the manufacturer. Eighty percent confluent cells were washed twice with free-serum medium and the mix of FuGene and plasmid DNA was added. In every experimental condition, 6 to 8 μg of plasmid DNA per Petri (10 $cm^2$) were used. Cells were grown for 48 h posttransfection before use. In co-transfection experiments, half amount of each plasmid DNA was used.

HEK293 (Human Embryonic Kidney, ECACC 96121229 from Sigma-Aldrich Chemie GmbH) were cultured in DMEM (Dulbecco's Modified Eagle Medium PAN biotech, Aidenbach, Germany), supplemented with 4.5 g/L glucose, 2 mM L-glutamine, 10% FBS, 100 units/ml penicillin, and 0.1 mg/ml streptomycin and maintained at 37°C and 5% $CO_2$. Cells cultured in 15-cm dishes were split at a 1:36 ratio into 6-well plates containing poly-D-lysine (PDL)-coated 24 mm glass coverslips. Cells were transfected using Effectene transfection reagent (QIAGEN) according to the manufacturer's instructions. Cells seeded on PDL-coated coverslips in 6-well plates were transfected 16 h after seeding with 0.6 μg plasmid/well.

## Fluorescence microscopy

Fluorescence microscopy experiments were performed either on a Leica SP5 (or SP8) Confocal Microscopes, using HyD photon counting detectors and an Argon/Ion laser (or white light laser) source for 488 nm/633 nm excitation and a solid-state diode laser (or white light laser) for 561 nm excitation. A solid-state diode laser was further used for 405 nm excitation. A 40× 1.3 NA objective was used, and the electronic zoom was set to achieve a pixel size of 50 nm. For visualization, mitochondria of transfected cells were labeled for 30 min at 37°C with 50 nM Mitotracker Deep Red FM (Invitrogen, Thermo Scientific).

## Ligand binding studies

Cells were lysed 48 h posttransfection with ice-cold 2 mM EDTA, 1 nM Na-HEPES (pH 7.4), scraped with a rubber policeman in the presence of approximately 4 ml rinsing medium, and homogenized with a Polytron homogenizer (ULTRA-TURRAX) in binding assay buffer (50 mM Tris-HCl (pH 7.4), 155 mM NaCl, 0.01 mg/ml bovine serum albumin). Radioligand bindings were carried out at 37°C for 3 h in a final volume of 1 ml. Nonspecific binding was determined in presence of 1 μM atropine. For competition binding assays, increasing concentrations of carbachol (from 0.01 μM up to 10 μM) were used in presence of 200 pM [$^3$H]N-Methylscopolamine ([$^3$H]NMS; 78.9 Ci/mmol; PerkinElmer).

The bound ligand was separated from the unbound ligand using glass-fiber filters (Whatmann, GF/B) with a Brandel Cell Harvester, and filters were counted with a scintillation β-

counter. $K_D$ values of [$^3$H]NMS were calculated in direct saturation experiments, whereas carbachol Inibitory Concentration (IC)$_{50}$ values were calculated in competition curves, against 400 pM [$^3$H]NMS, fitted to one binding models using the iterative, nonlinear least-squares regression analysis of OriginPro 7.5 software (OriginLab Corporation).

## Phosphatidylinositol breakdown assay

Transfected COS-7 cells were incubated with myo-[$^3$H]inositol (23 Ci/mmol; PerkinElmer) for 48 h. Immediately before the experiment, the cells were washed twice with PBS and incubated for 15 min in Eagle's medium containing 10 mM LiCl and 20 mM HEPES. The medium was then replaced by 0.25 ml of the same medium containing the experimental agents. After a 1-h incubation at 25°C, the reaction was arrested by the addition of 0.75 ml of 7.5% (w/v) ice-cold trichloroacetic acid followed by a 30-min incubation on ice. The trichloroacetic acid was extracted with water-saturated diethyl ether (3 × 4 ml), and levels of IP$_1$ determined by anion exchange chromatography.

## Adenyl cyclase assay

COS-7 cells were transfected with plasmid(s) containing the receptor of interest plus adenyl cyclase V, and 24 h after transfection, the cells were detached with trypsin and re-cultured in 24-well plates. Following an additional period of 24 h culture, they were assayed for adenyl cyclase activity. The assay was performed in triplicate. In brief, cells were incubated in 24-well plates for 2 h with 0.25 ml/well of fresh growth medium containing 5 μCi/ml [$^3$H]adenine (PerkinElmer). The medium was then replaced with 0.5 ml/well of DMEM containing 20 mM HEPES (pH 7.4), 0.1 mg of bovine serum albumin, and the phosphodiesterase inhibitors 1-methyl-3-isobutylxanthine (0.5 mM) and Ro-20–1724 (0.5 mM). AC activity was stimulated by the addition of 1 μM forskolin in the presence or absence of carbachol. After 10 min of incubation at 30°C, the medium was removed, and the reaction terminated by addition of perchloric acid containing 0.1 mM unlabeled cAMP followed by neutralization with KOH. The amount of [$^3$H]cAMP formed under these conditions was determined by a two-step column separation procedure [3].

## Immunoelectron microscopy (IEM)

Samples were fixed for 3 h at 4°C with a mixture of 4% PFA and 0.05% GA in 0.1M PB (pH 7.4). After rinsing in the same buffer, samples were dehydrated in a graded series of ethanol and embedded in medium grade LR White resin, following addition of the LR White accelerator (London Resin Company, Berkshire, England). The resin was then polymerized for 20 min at room temperature in tightly capped gelatine capsules. Ultrathin sections were obtained using a Reichert Ultracut ultramicrotome equipped with a diamond knife and collected on nickel grids. For immunogold labeling, 60 to 80 nm thick sections were processed following the incubation protocol originally developed by Aurion (Wageningen, the Netherlands). Unspecific binding was prevented by treating sections for 20 min with 0.05 M glycine in PBS, at pH 7.4. A subsequent block step was made by washing sections for 30 min in 5% BSA, 5% normal goat serum, and 0.1% cold water fish skin gelatine. Sections were soaked overnight 4°C in incubation buffer (PBS and 0.1% BSA-cTM (pH 7.4)) (3 × 5 min) in a moist chamber containing the Myc-Tag (9B11) mouse antibody (Cell Signaling Technology) diluted 1:50 with incubation buffer. Sections were then washed in incubation buffer (6 × 5 min) and incubated for 1 h with a secondary goat anti-mouse antibody conjugated to 10 nm gold particles (EM Grade 10nm Goat anti-Mouse IgG Conjugate EM.GAM10/1, British BioCell International, United Kingdom), diluted 1:10 in incubation buffer. After rinsing in incubation buffer (6 × 5

min), grids were washed in PBS (6 × 5 min). Sections were subsequently stained with uranyl acetate and observed with a Jeol JEM EXII Transmission Electron Microscope at 100 kV. Micrographs were acquired by the Olympus SIS VELETA CCD camera equipped the iTEM software. Control sections were obtained by omitting the primary antibody from the incubation mixture.

## Isolation of yeast mitochondria

Mitochondria were isolated as in Kritsiligkou and colleagues [36]. In short, D273-10B yeast were grown to OD600 1 in yeast peptone lactate. Cells were pelleted by centrifugation (3,000g) and washed in ddH$_2$O and the cell pellet weight was calculated. The yeast cells were then washed in 100 mM Tris-SO$_4$ (pH 9.4), 10 mM DTT for 30 min in the presence of constant agitation (120 rpm). The cells were washed in sorbitol buffer (1.2 M sorbitol, 10 mM KPi (pH 7.4)) followed by resuspension in sorbitol buffer (5 ml/g of cells) supplemented with zymolyase 20T (3.5 mg/g of cells) and incubation for 45 min under constant agitation (120 rpm) to create spheroplasts. The spheroplasts created in this process were washed in sorbitol buffer and resuspended in breaking buffer pH 6.0 (BB pH 6.0) (600 mM sorbitol buffer, 20 mM MES-KOH (pH 6.0), 1 mM PMSF). The spheroplasts were then broken in BB pH 6.0 using a glass type B pestle homogenizer for 15 strokes on ice. The cell debris was removed via differential centrifugation; first low speed centrifugation (1,480g, 4˚C) then high speed (12,000g, 4˚C). Crude mitochondria were then purified further via gradient centrifugation on a 14.5% to 20% Nycodenz gradient and spun at 202,496 g at 4˚C. The pure mitochondria were collected with a 19-gauge needle, washed in BB pH 7.4 (600 mM sorbitol buffer, 20 mM HEPES-KOH (pH 7.4)) and protein concentration estimated. Finally, the mitochondria are frozen at −80˚C at a concentration of 20 mg/ml in BB pH 7.4 supplemented with 10 mg/ml fatty acid free BSA.

## Import of M$_2$ receptor and M$_2$tail(368–466) in isolated yeast mitochondria

The M$_2$ receptor and M$_2$tail(368–466) were cloned into the pSP64 in vitro translation vector and translated with a $^{35}$S radiolabeled methionine label using the TNT coupled reticulocyte lysate system (Promega) following manufacturers guidelines. Each import reaction comprised of 50 µg of isolated yeast mitochondria resuspended in 100 µl import buffer 1× (600 mM sorbitol, 2 mM KH$_2$PO$_4$, 50 mM KCl, 50 mM HEPES-KOH (pH 7.4), 10 mM MgCl$_2$, 2.5 mM Na$_2$EDTA (pH 7), 5 mM L-methionine, 1 mg/ml fatty acid free BSA) supplemented with 2 mM ATP and 2.5 mM NADH. Approximately 5 µl of TNT translated protein was added to each import reaction and the samples were incubated at 30˚C for 5, 10, or 30 min. Import was ceased by the transfer of samples on ice. Where indicated, samples were treated with 0.1 mg/ml Trypsin in breaking buffer (600 mM sorbitol, 20 mM HEPES-KOH (pH 7.4)) for 30 min on ice followed by protease inactivation with the addition of soybean trypsin inhibitor (SBTI) (1 mg/ml) for 10 min. Mitochondria were reisolated by centrifugation (16,000g, 4˚C) and resuspended in either 2× Laemmli sample buffer (65 mM Tris-HCl (pH 6.8), 25% [v/v] glycerol, 2% SDS, 0.01% bromophenol blue) for SDS-PAGE analysis or solubilization buffer (0.1 mM EDTA, 50 mM NaCl, 10% [v/v] glycerol, 20 mM Tris-HCl (pH 7.4), 1% [w/v] digitonin) for Blue Native-PAGE analysis. For NaCO$_3$ extraction, 100 µg of mitochondria were pelleted after SBTI at 16,000g, 4˚C and resuspended in 200 µl of 0.1 M NaCO$_3$ for 30 min on ice followed by ultracentrifugation (100,000 g, 30 min, 4˚C). Supernatants were TCA-precipitated (10% [v/v] final) for 20 min on ice followed by centrifugation (16,000g, 4˚C, 20 min). The TCA precipitated supernatant and the pellet after ultracentrifugation were resuspended in 2× Laemmli sample buffer and proteins were separated by SDS-PAGE.

### Ethics statement

All animal experiments were carried out according to the German animal welfare act, considering the guidelines of the National Institute of Health and the 2010/63/EU Directive of the European Parliament on the protection of animals used for scientific purposes. Animal experiments were conducted with the approval of the Landesamt fûr Gesundheit und Soziales (LAGeSo, Berlin) under the internal sacrification license X 9010/18, under the institute (MDC) licence X9014/11.

The animals had free access to food and water and were kept in individually ventilated cages under a 12h:12h light/dark regime (light from 6:30 AM to 6:30 PM), a constant $22 \pm 2°C$ temperature, and $55 \pm 10\%$ humidity.

The transgenic line B6.Cg-Chrm2tm1.1Hze/J, also known as Chrm2-tdT-D knock-in, was obtained from "The Jackson Laboratory" (Stock No: 030330). They are heterozygous animals, designed to have expression of a Chrm2/tdTomato fusion protein directed by the endogenous $M_2$ muscarinic acetylcholine receptor promoter/enhancer sequences.

### Tissue preparation and immunoblot of endogenous $M_2$ receptor in Chrm2-tdT-D knock-in mice

Mouse tissues were weighed out and ice-cold lysis buffer was added to approx. 0.2 to 0.5 mg tissue/ml buffer (10 mM $K_2HPO_4$, 150 mM NaCl, 5 mM EDTA, 5 mM EGTA, 1% Triton X-100, 0.2% Na-deoxycholate). Fresh protease inhibitors PMSF 0.5 mM, $Na_3VO_4$ 0.1 mM, NaF 50mM, Jenny's mix (SBTI 3.2 µg/ml, Aprotinin 2 µg/ml, Benzamidin 0.5 mM) were added to the buffer.

Samples were first homogenized, and then pellet debris were centrifuged at 4°C for 30 min at max speed (tabletop) and the supernatant (soluble proteins) was transferred into fresh tube (pellets = non-soluble proteins).

Protein concentrations were determined with BCA assay. Cell lysates were loaded using Laemmli Buffer 2x (Sigma Aldrich) in equal mass (see above for the immunoblot procedure) into 8% polyacrylamide gels.

The membrane was then incubated overnight at 4°C with primary antibody solution (1:1,000 anti β-actin (13E5) rabbit primary antibody #4970 from Cell Signaling) (1:1,000 anti GAPDH) (1:1,000 anti-tdTomato rabbit primary antibody #632496 from Takara).

### Isolation of mitochondrial fractions and immunodetection

HEK293 cells were seeded in 15 cm plates and transfected after 24 h according to manufacturer protocols (Effectene, Qiagen) using 5 µg of plasmid DNA, and 48 h after transfection, cells were harvested and centrifuged at 1,000 rpm for 10 min, resuspended in NaCl 0.9% (w/v), and centrifuged at 1,000 rpm for 10 min.

Mouse tissues were freshly excised and placed in ice-cold PBS (1×). The samples were washed with NaCl 0.9% (w/v). The homogenization steps were performed with Precellys Evolution Homogenizer at 4,500 rpm for 10 s.

The samples were processed according to manufacturer protocols (Qproteome Mitochondria Isolation Kit, Qiagen). From the different steps were obtained 3 different fractions: the cytosolic, the microsomal, and the mitochondrial fraction.

The fresh mitochondrial pellet was resuspended in 150 µl of ice-cold lysis buffer and the protein concentration of the 3 fractions from each sample was quantified with a BCA assay (see above).

Cell lysates were loaded into 12% polyacrylamide gels using an SDS loading buffer (6× solution: 2.28 g Tris-HCl, 2.313 DTT, 6 g SDS, 30% Glycerol, and 0.03% bromophenol blue in a final volume of 50 ml) in equal mass (see above for the immunoblot procedure).

The membrane was then incubated overnight at 4˚C with primary antibody solution using the following concentrations: 1:1,000 anti β-actin (13E5) rabbit primary antibody (#4970 from Cell Signaling), 1:1,000 anti-tdtomato rabbit primary antibody (#632496 from Takara), 1:1,000 anti β-tubulin mouse primary antibody, 1:1,000 anti GAPDH mouse primary antibody, 1:1,000 anti COXIV mouse primary antibody (#ab33985 from abcam).

## Quantification of M₂tail(368–466) mitochondrial import kinetics from confocal time sequences

Before imaging, $2.5 \times 10^4$ HEK-TSA cells were seeded in each PDL-coated well of the µ-Slide 8 well (Ibidi) with DMEM (ScienCell) supplemented with 10% FBS and 1% P/S overnight, and transfected with a total of 2 µg C-terminal EGFP tagged M₂ tail using Lipofectamine 2000 (Thermo Fisher Scientific) according to the manufacturer protocol after washing with 200 µl PBS and refreshing with 100 µl DMEM (ScienCell) supplemented with 10% FBS and 1% P/S. After transfection for 5 h, mitochondria of transfected cells were rapidly labeled with 50 nM Mitotracker Deep Red FM (Invitrogen, Thermo Scientific) in FluoroBrite DMEM medium (Thermo Scientific) supplemented with 1% P/S for 20 min after the removal of old culture medium. After mitochondria labeling, the µ-Slide were washed twice with FluoroBrite DMEM medium (Thermo Scientific) and subsequently mounted onto the TCS SP8 confocal microscope (Leica) (see imaging). Firstly, the stage of the microscope was equilibrated in temperature, and the acquisition mode was set to the tile scanning. After storing the position of each desired transfected cells, the imaging mode was set to XYT and an HC PL APO CS2 40×/ NA1.3 oil immersion objective was selected. Both DIC channel and HyD were utilized to obtain 512 × 512 pixels images. Laser wavelengths were set as described in the section Fluorescence Microscopy above. All settings remained the same throughout the whole measurement. Images were taken at every 30 min over a 5 h interval. The images were afterwards analyzed by ImageJ software, using the Plugin "JACoP." Briefly, the degree of colocalization between EGFP-tagged M₂tail(368–466) (CH1) and Mitotracker stained mitochondria (CH2) was quantified by Manders colocalization coefficients [38]. The 2 colocalization coefficients $MCC1$ ($MCC1 = \frac{CHB}{CH1}$) and $MCC2$ ($MCC2 = \frac{CHB}{CH2}$), CHB indicates overlapping pixels in channel 1 and channel 2. A threshold was determined for individual channel to distinguish specific signals from nonspecific ones. Due to their sensitivities to thresholds, the Pearson correlation coefficient (PCC) [15,38], as well as the faster decaying attribute of nonspecific signals, were also taken into account while estimating thresholds for each channel. For the analysis of each cell, the threshold for Mitotracker channel was kept constant, whereas the threshold for EGFP channel was evaluated based on the PCC value. $MCC2$ (fraction of M₂tail inside mitochondria) was displayed.

## Transfection and expression of split-GFP constructs in HEK293 cells

To perform the sequential split GFP complementation assay illustrated in **Figs 4D, 5C and 5D**, $3 \times 10^5$ HEK-TSA cells were seeded on a 10% poly-D-lysine (PDL)-coated 25 mm coverslip in each well of a Falcon 6-well flat-bottom plate (StemCell) and cultured overnight. Cells were transfected using Effectene (see above) with a 0.4 µg Mito-GFP1-10 plasmid. The cells were then washed once with fresh 2 ml PBS first and supplemented with 1.5 ml fresh DMEM medium (ScienCell) supplemented with 10% FBS and 1% P/S. After adding the transfection mixture, the cells were cultured overnight. After the first transfection, a second transfection of

a 0.6 μg plasmid encoding either the C-terminal GFP11 tagged $M_2$tail(368–466) or $M_2$ receptor was performed using the Effectene Transfection Reagent (Qiagen). After washing each well with 2 ml PBS, the old medium was exchanged with 1.5 ml low glucose Minimum Essential Medium (MEM) (Gibco, Thermo Scientific) supplemented with 1% P/S, in order to create a stress-like cell starvation condition. After a 4 h starvation, the cells were fixed using 4% poly-formaldehyde (PFA)/PBS at 37˚C for 30 min and washed twice with PBS before imaging.

## Oxygen consumption measured with the Seahorse assay

$5 \times 10^4$ COS-7 cells were seeded in seahorse 24MW plates with Complete media (DMEM supplemented with 10% FBS and 1% P/S) and then incubated for about 4 h at 37˚C in a cell incubator. After this incubation, cells were washed twice with Seahorse media supplemented with glutamate (1 mM), pyruvate (1 mM), and glucose (20 mM) and then incubated for 1 h at 37˚C in an incubator without $CO_2$ before being analyzed in the Seahorse XF Analyzers. In particular, the baseline cell OCR was measured at the beginning of the assay, followed by measurements of ATP linked respiration and maximal respiratory capacity after the addition of respectively oligomycine (1 μM) and the FCCP protonophore (1 μM). Finally, rotenone (1 μM), an inhibitor of complex I, and antimycin (1 μM), inhibitor of complex III, were added to shut down completely the electron transport chain (ETC) function, revealing the non-mitochondrial respiration.

## Extracellular oxygen consumption

HEK293 cells 24 h after transfection and hPSCs were harvested and seeded respectively in 96 multiwell black plate and pre-coated (Geltrex A1413302, Thermo Fisher) 96 multiwell black plate, $5 \times 10^4$ cells for well; after incubation overnight at 37˚C and in 5% $CO_2$, the cell culture medium was replaced by HBSS and the assay were performed according to the manufacturer protocols (abcam Extracellular Oxygen Consumption Assay).

## Generation of M368A hiPSC clone

The hiPSC line BIHi005-A (https://hpscreg.eu/cell-line/BIHi005-A) was used to generate an isogenic iPSC clone (BIHi005-A-39) with M368A mutation using a protocol previously described [39]. In brief, small guide RNA (sgRNA) targeting the $M_2$ gene close to the mutation site to be corrected were designed using the http://crispor.tefor.net/ webpage and the sgRNA (5′- AGTCATCTTCACAATCTTGC-3′) were synthesized by Integrated DNA technologies (https://eu.idtdna.com). The donor ssODN template was designed to convert the ATG to GCT and was synthesized as an Ultramar DNA oligo by IDT. (5′- AAAAGTGACTCATGTACCC CAACTAATACCACCGTGGAGGTAGTGGGGTCTTCAGGTCAGAATGGAGATGAA AAGCAGAATATTGTAGCCCGCAAGATTGTGAAGGCTACTAAGCAGCCTGCAAA A-3′). The Ribonucleoprotein (RNP) complexes were prepared by incubating of 1.5 μg Cas-9 protein and 360 ng gRNA for 10 min at room temperature. For delivery of RNPs and ssODN template, 10 μl cell suspension containing Cas9 RNPs and ssODN and $1 \times 10^5$ cells were electroporated using the Neon transfection System (Thermo Fisher Scientific). The electroporated cells were plated in one well of 6 well plate with StemFlex media (Thermo Fisher Scientific) supplemented CloneR (Stemcell Technologies). Three days after the transfection, we analyzed the bulk population using the Sanger sequencing to estimate the editing efficiency. Thereafter, the automated single-cell cloning of the genome-edited cell pool was performed as described in protocol [39]. The clones were screened by SANGER sequencing. The positive confirmed clones were banked and characterized for pluripotency and karyotype stability as described in [40].

Briefly, for pluripotency assay, single cells were labeled with conjugated antibodies. Surface markers (SSEA1, SSEA4, and TRA1-60) staining was performed in unfixed cells by incubation with antibodies diluted in 0.5% BSA/PBS (10 min, 4˚C, dark). Intracellular markers (OCT3/4, NANOG) were stained by fixation/permeabilization solution (Miltenyi Biotec, 130-093-142) (30 min, 4˚C, dark) followed by antibody incubation in permeabilization buffer (Miltenyi Biotec) (30 min, 4˚C, dark). Analysis was done using the MACSQuant AnalyzerVYB and FlowJo v10.4.

Briefly, for karyotype stability, genomic DNA was isolated using the DNeasy blood and tissue kit (Qiagen, Valencia, California, United States of America) and samples were analyzed using the human Illumina OMNI-EXPRESS-8v1.6 BeadChip. First, the genotyping was analyzed using GenomeStudio 1 genotyping module (Illumina). Thereafter, KaryoStudio 1.3 (Illumina) was used to perform automatic normalization and to identify genomic aberrations utilizing settings to generate B-allele frequency and smoothened Log R ratio plots for detected regions. The stringency parameters used to detect copy number variations (CNVs) were set to 75 kb (loss), 100 kb (gain), and CN-LOH (loss of heterozygosity) regions larger than 3 MB.

## Culture and handling of hiPSCs cells

hiPSCs were cultured on Geltrex (Thermo Fisher) coated plates using: Essential 8 medium (Thermo Fisher). The culture was routinely monitored for mycoplasma contamination using a PCR assay, and 10 μm ROCK Inhibitor Y-27632 2HCl (Selleck Chemicals, #SEL-S1049) was added after cell splitting to promote survival. hiPSC cultures were kept in a humidified atmosphere of 5% $CO_2$ at 37˚C and 5% oxygen. hiPSCs were undirectedly differentiated to a mixture of 3 lineages (ectodermal, endodermal, and mesodermal) by replacing Essential 8 medium with DMEM-F12 medium supplemented with 10% FBS for 7 days, and 10 μM ROCK Inhibitor Y-27632 2HCl (Selleck Chemicals) was added after cell splitting to promote survival.

## Cell proliferation assay with HEK293 cells

$4 \times 10^5$ HEK293 cells 24 h after transfection and $4 \times 10^5$ hPSCs were seeded respectively in T25 flask and pre-coated T25 flask, in culture medium with atropine at final concentration of 100 nM; the medium with atropine was replaced every day. After 72 h from transfection of the HEK293 and from seeding of the hiPSCs, the medium and the cells were collected and centrifuged, the pellet was resuspended in the same amount of medium for each pellet; an aliquot of the resuspension was mixed with the same amount of Trypan blue (ratio 1:1), 10 μl were used for counting in a Neubauer chamber.

## Fluorescence plate reader assays

HEK293 cells were grown on 6 well plates without coverslips, and, upon reaching 60% to 70% confluence, the cells were transfected using Effectene (Quiagen, Venlo Netherlands) according to the manufacturer's instructions. After 24 h, the cells were split and transferred to 96 well (black plate) at a seeding density of 50,000 cells/well. The 96 well plate was pre-coated with Poly-D-Lysine for 30 min at RT. The medium was removed and each well was washed 3× with 100 μl HBSS. The cells were finally analyzed by a fluorescence plate reader (Neo2, BioTek, Bad Friedrichshall) in HBSS.

## Statistical analysis

Significance values reported in the graphs, in comparing tabulated values, were determined by a one-tailed Student $t$ test using Prism 9 (GraphPad Software), and significance values ($p$-

values) are indicated according to the following legend: **** $p < 0.0001$; *** $0.0001 < p < 0.001$; ** $0.001 < p < 0.01$; * $0.01 < p < 0.05$. $p$-Values larger than 0.05 are deemed nonsignificant. Unless otherwise indicated, bar charts and scatter plots indicate average value with SEM.

## Supporting information

**S1 Fig. Snake diagram of the $M_2$ receptor.** Key residues are highlighted [41]. Asterisks in the red circles denote the insertion of stop codons within transmembrane regions V (196) and VI (400), as well as the i3 loop (228). The in-frame methionine highlighted in yellow are the ones within the third loop that have been substituted with stop codons to investigate the initiation start site of $M_2$ C terminal fragment. Adapted from GPCRdb.
(AI)

**S2 Fig. $M_2$ receptor mutants: signaling, alternative splicing, and western blotting.** (**A**) Dose response curve of carbachol induced ERK phosphorylation in HeLa cells transiently transfected with $M_2$ and $M_2$stop228 muscarinic receptors. Cells were stimulated for 5 min with the indicated concentration of carbachol. Phosphorylated ERK (P-ERK) was normalized vs. total ERK (Tot ERK). Significance values reported in the graphs were determined by a one-tailed Student $t$ test, $p$-values: ** $0.001 < p < 0.01$; * $0.01 < p < 0.05$ was calculated against carbachol 0 μm. (**B**) mRNAs were extracted by COS-7 cells transfected with $M_2$ wild type and $M_2$stop228, subjected to reverse transcriptase and the resulting cDNA amplified by PCR with 2 oligos directed to the 5′ and 3′ end of the receptors. The gel shows, in both $M_2$ wild type and $M_2$stop228, a single band of 1,401 bp corresponding to the full-size receptor, running at the same level of the band amplified directly from the $M_2$ pcD plasmid. (**C**) Western blot of whole cell lysates of HEK293 cells. $M_2$-Myc and $M_2$tail(368–466)-Myc were transiently transfected in HEK293 cells and immunodetected via western blot together with an untransfected control. Loading control was verified by immunoblotting tubulin. (**D**) Western blot of whole-cell lysates untransfected HEK293 (lane 1) and then HEK293 cells transfected with myc-$M_2$Stop228 (lane 2) and myc-$M_2$Trunk(1–228) (lane 3). Loading control was verified by detecting tubulin. (**E**) Western blot of whole cell lysates of COS-7 cells. $M_2$-Myc, $M_2$stop228-Myc, $M_2$tail(368–466)-Myc, $M_2$stop400-Myc were transiently transfected in COS-7 cells and immunodetected via western blot together with an untransfected control. Source data for panel A can be found in S1 Data.
(AI)

**S3 Fig. Schematic representation of wild-type muscarinic $M_2$ (human) receptors and derived mutants.** Each construct was obtained as described under **Materials and methods** and **S1 Methods**. The left column indicates the mRNA product from each construct, the right column the expected protein product. For the last 2 constructs, above mRNA product, and below the expected protein product.
(AI)

**S4 Fig. Expression of the bicistronic plasmid Sirius-$M_2$i3(417n)-EGFP in 4 different cell lines, ribosome profiling data, and sequence alignment.** (**A**) Expression of Sirius was driven by the canonical scanning mechanism of translation initiation, while EGFP expression was driven by an IRES-dependent mechanism. All Sirius-$M_2$i3(417n)-EGFP transfected cells expressed both the Sirius and EGFP proteins. On average, 25 ± 4% of the COS-7 cells transfected with Sirius-$M_2$i3(417n)-EGFP plasmid were blue fluorescence positive, while 17 ± 2% were green fluorescent positive, with only a few cells being only green. Excitation conducted at 405 nm (410–450 nm detection) for Sirius, and at 488 nm (500–550 nm detection) for GFP. Scale bars are 10 μm throughout. (**B**) Ribosome profiling data from heart-specific

transcriptomic databases [34], where the $M_2$ receptor is highly expressed. When looking at left ventricle data, the ribosome coverage data is prominent in correspondence of the third i3 loop in frame-methionine M368 (highlighted by the red arrow). In this case, also p-site hits are observed. (**C**) Alignment of the nucleotide sequence 1072–1104 of the $M_2$ i3 loop with analogous sequences of the other 4 muscarinic receptors. In yellow are highlighted the conserved nucleotides. In bold are the in frame ATG codons. Next to each sequence is indicated the codon number of the in-frame ATG and its distance in nucleotides from the beginning of TM domain 6.
(EPS)

**S5 Fig. Expression of the bicistronic plasmid Sirius-M₃i3(558n)-EGFP.** (**A**) Schematic representation of the bicistronic plasmid bearing the i3 loop of the rat muscarinic $M_3$ receptor (558 nucleotides from nucleotide 880 to nucleotide 1437) between the coding regions of the Sirius and EGFP fluorescent proteins. (**B**) EGFP expression in COS-7 cells transfected with the bicistronic plasmid Sirius-M₃i3(558n)-EGFP. Source data for panel B can be found in S1 Data.
(AI)

**S6 Fig. Subcellular localization of the M₂tail(368–466).** (**A**) Confocal image displaying the colocalization (left of $M_2$tail(368–466)-EGFP (green, middle) in HEK293 cells, co-expressed with the outer mitochondrial membrane marker Tomm20-mCherry (red, right). (**B**) Expression of $M_2$tail-EGFP displaying marked mitochondria and cytosolic localization. (**C**) Glasbey colorscale representation of panel (B) highlighting the low-intensity pixels. (**D**) autocorrelation function of the $M_2$tail(368–466)-EGFP diffusion taken at or in proximity of the basal membrane, yielding a mean diffusion coefficient of 0.11 $\mu M_2$/s, incompatible with membrane diffusion. (**E**) Co-expression of $M_2$stop228 and $M_2$tail(368–466)-EGFP. $M_2$stop228 is unlabeled. $M_2$tail(368–466)-EGFP maintains the localization to the mitochondria observed in cells expressing $M_2$tail(368–466)-EGFP alone, but it is also localized to the plasma membrane, visible in panel (**F**) Glasbey colorscale representation of panel E highlighting the low-intensity pixels. (**G**) Autocorrelation function of the $M_2$tail(368–466)-EGFP diffusion taken at or in proximity of the basal membrane, yielding a mean diffusion coefficient of 0.013 $\mu M_2$/s, compatible with membrane diffusion. (**H**) Cellular localization of $M_2$tail-mRuby2, together with Mitotracker deep red staining of the mitochondrial network and corresponding DIC image. (**I**) Cellular localization of $M_2$tail-tdtomato and corresponding DIC image. (**J**) Confocal sequential images displaying the localization of fluorescently labeled $M_2$tail(368–466)-EGFP (green), together with the mitochondrial network (magenta), colocalization (white) and DIC image in COS-7 cells. All imaging panels originate from Laser Scanning Confocal Microscope sequential acquisitions, with laser lines 488 nm (EGFP) 561 nm (mRuby2, mCherry and tdtomato) and 633 nm (Mitotracker deep red), and corresponding emission filters in the ranges 520–600 nm, 570–620 nm and 640–750 nm. Scale bars are 10 μm. **K**, Western blot (10% TRIS-Glycine PAA gel) of the cytosolic and microsomal fractions resulting from the mitochondria purification of lysates of HEK293 transfected with myc-tagged constructs, as displayed in **Fig 3C**. Loading control involved immunoblotting for β-actin. Source data for panels D and G can be found in S1 Data.
(EPS)

**S7 Fig. Mitochondrial localization of the M₂tail fragment under endogenous IRES production and in vitro mitochondrial import of M₂tail(368–466). A,** Confocal image of $M_2$-EGFP (green) expressed in conjunction with Mitotracker (magenta) in HEK293 cells. Separate panels below. **B,** Confocal image of HEK293 cell co-expressing $M_2$tail(368–460) and $M_2$fr.sh-mRuby2, together with the merged fluorescence image and DIC. **C,** Confocal micrographs of

representative H9c2 cells expressing $M_2$wt-EGFP, $M_2$tail(368–466)-EGFP, $M_2$fr.sh. M368A-EGFP, $M_2$M368A-EGFP, $M_2$stop228-EGFP and an untransfected control. Panels display, from top to bottom EGFP (green), Mitotracker (Magenta), overlay (EGFP, Mitotracker and, where present, Hoechst 33342 (Cyan)) and DIC (grays). Scale bars are 10 μm. Confocal sequential acquisitions were performed with 405 nm excitation and 420–460 nm detection (Hoechst), 488 nm excitation and 520–600 nm detection (EGFP), and 633 nm excitation and 650–750 nm detection (Mitotracker) using HyD detectors in Photon Counting Mode. Manders Correlation Coefficient $M_2$ (fraction of green/$M_2$tail features within magenta/mitochondria features) is indicated on the overlay images. **D,** $^{35}$S-labeled $M_2$tail(368–466) or AAC (an integral internal mitochondrial membrane protein) were imported into isolated yeast mitochondria originating from different knock-out lines for outer membrane transporters (ΔTom). ΔTom5 = 68% of WT, ΔTom6 = 87% of WT, ΔTom7 = 87% of WT, ΔTom70 = 59% of WT. Samples were incubated for 30 minutes, and then washed in breaking buffer. After import, 100 μg of mitochondria were subject to Carbonate extraction to determine if $M_2$tail (368–466) is integrated into a lipid bilayer (Pellet, 68%) or loosely associated/soluble (Supernatant, 32%). Samples were then loaded onto a 12% Tris-Tricine gel followed by semi-dry transfer and visualized via a phosphorimager (n = 1). To test for equal mitochondrial loading anti-αPorin was used via western blot. (right)$^{35}$S-labeled $M_2$tail(368–466) and αPorin were imported into isolated wild-type or ΔTom70 mitochondria for the indicated timepoints. Samples were then washed in breaking buffer and loaded on blue-native page followed by gel drying and visualized via a phosphorimager (n = 1). **E,** Representative IEM micrographs of a control (left) and transfected (right) COS-7 cells. Scale bar is 1 μm. Statistics of IEM staining, both in transfected as well as control cells. The discernible 'dark spots' observed in the control, likely attributed to varied exposure settings during imaging, exhibit a noticeably larger size and less distinct shapes compared to the transfected samples, where circular, sharp, and intense gold nanoparticles are evident. Source data for panel E can be found in S1 Data. (AI)

**S8 Fig. Cellular localization of split GFP constructs in HEK293 cells. A,** Confocal images displaying the mitochondrial localization of $M_2$tail(368–466)-GFP11 and mito-GFP1-10. **B,** co-expression of SMAC-GFP11 and SMAC-GFP1-10 (positive control) and corresponding Mitotracker deep red image. **C,** $M_2$tail-GFP11 + $M_2$tail-GFP1-10, with corresponding DIC image. **D,** mito-GFP11 and mito-GFP1-10 (positive control). Two representative experiments are displayed in panels A-D out of n = 5 transfections for each condition. **E,** $M_2$tail(368–466)-GFP11 and SMAC-GFP1-10. **F,** mito-GFP1-10 alone (negative control). One representative experiment is shown in panels E-F out of n = 4 transfections for each condition. EGFP (green) was excited at 488 nm and fluorescence collected between 500–600 nm. Mitotracker deep red (magenta) was excited at 633 nm and fluorescence collected between 650–750 nm. Scale bars are 10 μm. (AI)

**S9 Fig. Mitochondrial localization of the $M_2$tail(368–466) fragment upon IRES production. A-E,** Mitochondrial localization of the IRES mediated C-terminal fragment is enhanced upon cellular stress. HEK293 cells transfected with the super construct $M_2$-mRuby2-S-TOP-$M_2$-i3-tail-EGFP display increased localization of the EGFP labeled portion to the mitochondria, upon serum starvation for 2 hours and incubation of the cells in HBSS buffer, in presence of Mitotracker: **A,** is IRES-driven $M_2$tail(368–466)-EGFP. **B,** Cap dependent $M_2$-mRuby2 + IRES-driven $M_2$tail(368–466)-mRuby2, both derived from the $M_2$-mRuby2 gene. **C,** Mitotracker Deep Red **D,** Overlay of Mitotracker Deep Red and mRuby channel (colocalization in magenta). Zoom-in of yellow square immediately below the panel. **E,** Overlay

between GFP and mRuby2 (colocalization in yellow). **F-L**, Localization of the mutant construct $M_2$(M368A)-mRuby2-STOP-$M_2$-i3-tail-EGFP in HEK293 cells after 2 hours incubation in HBSS buffer. The same legend as in **A**-**E** applies. Bottom: Schematic structure of the mega constructs $M_2$(M368A)-mRuby2-STOP-$M_2$-i3-tail-EGFP. This construct is similar to the $M_2$-mRuby2-STOP-$M_2$-i3-tail-EGFP construct but the methionine 368 in the $M_2$ sequence has been replaced with alanine. The asterisks represent the stop codons at the end of mRuby2 and EGFP. EGFP was excited at 488 nm and fluorescence collected between 500–600 nm; mRuby2 was excited at 561 nm and fluorescence collected between 580–620 nm. Mitotracker deep red was excited at 633 nm and fluorescence collected between 650–750 nm. Scale bars are 10 μm.

(AI)

**S10 Fig. Supplemental data on the effect of the $M_2$tail(368–466) fragment on mitochondrial function and cell metabolism. A,** Representative experiment of Seahorse assay showing the time course of oxygen consumption inhibition by $M_2$tail(368–466)-EGFP and wild type $M_2$-EGFP, compared to the same experiments performed in control cells transfected with EGFP. The rate of ATP-linked respiration measured after oligomycin administration is decreased in $M_2$tail and slightly decreased in $M_2$ cells as well as compared to the controls. FCCP, an ionophore, was added to each cell type to investigate the cellular bioenergetic reserves that seem to be slightly reduced only in the $M_2$ tail cells. **B,** Oxygen consumption measured by a Clark type electrode-based-polarographic method at a constant temperature of 37˚C. After 15 minutes of basal respiration, the ATP synthase inhibitor oligomycin was added at a concentration of 17 nM. Oxygen consumption in the absence and presence of oligomycin is represented by black and red bars, respectively. The blue bars represent the percentage of mitochondrial oxygen consumption (total minus oligomycin treated) in cells transfected with each construct. Significance values reported in the graphs were determined by a one-tailed Student t-test, p-values: ** $0.001 < p < 0.01$. **C,** is a representative experiment showing the time course of inhibition of oxygen consumption by $M_2$tail(368–466), compared to the same experiments performed in control cells and cells transfected with the $M_2$ wild type construct, upon addition of oligomycin. Results are given as ng-atom of oxygen/min/$10^6$ cells. **D,** Western blot of caspase-3 cleavage in COS-7 cells transfected with the indicated $M_2$ receptor mutants 48 h after transfection (top of panel, normal contrast; bottom of panel, increased contrast). Below, as a positive cleavage control, the blot for caspase-3 with and without Staurosporine incubation. **E,** Confocal sequential images of $M_2$tail(368–466)-EGFP (green), Mitotracker (magenta) and overlay upon 2 h of serum starvation. Scale Bars are 10 μm. **F,** Extracellular oxygen consumption assay of HEK293 cells transfected with the indicated $M_2$ receptor mutants using a fluorogenic dye quenched by Oxygen. Blank indicates the dye lifetime measured in wells containing the dye, but no cells. **G,** Normalized dose response to Carbachol stimulation of HeLa cells, comparing $M_2$ wild type to the $M_2$ M368A mutant. **H,** Normalized dose response curves based on FRET Gi biosensor [42], reflecting Galphai3 activation by two $M_2$ receptor mutants ($M_2$ wild type n = 7 transfections, $M_2$ M368A n = 3 transfections) stimulated by acethylcholine (AcOH) in HEK293 cells. Source data for panels A, B, C, F, G and H can be found in S1 Data.

(AI)

**S1 Table. Ligand binding and functional properties of $M_2$ receptor mutants with single and double stop codons.** Number of [3H]NMS binding sites for the indicated mutants. Untransfected COS-7 cells did not show any [3H]NMS specific binding. N.B. = no specific [3H]NMS binding.

(XLSX)

**S2 Table. Radioligand binding and activation properties of $M_3$ receptor mutants.** Binding characteristics and stimulation of phosphatidylinositol accumulation by $M_3$, $M_3$stop273 and the co-transfected $M_3$trunk(1–272) and $M_3$tail(M-388-586). Untransfected COS-7 cells did not show any [$^3$H]NMS specific binding. N.B. = no specific [$^3$H]NMS binding. (XLSX)

**S3 Table. Ligand binding of $M_3$ receptor mutants with single and double stop codons.** Number of [$^3$H]NMS binding sites in COS-7 cells transiently transfected with the indicated $M_3$ receptor mutants with single and double stop codons. (XLSX)

**S1 Data.** This folder contains the source data for Figs 2A, 2C, 4B, 5A, 5B, 6A–6D, 7D, 7E, S2A, S5B, S6D, S6G, S7E, S10A–S10C and S10F–S10H. (ZIP)

**S1 Raw Images.** This file contains the source images for all gels and Western blots reported in the manuscript. (PDF)

**S1 Text.** This files contains supplementary discussion related to some of the results. (DOCX)

**S1 Methods.** This file contains supplemntary methods. (DOCX)

## Acknowledgments

We are grateful to Dr. Marco Marcia (EMBL) for insightful discussion.

## Author Contributions

**Conceptualization:** Martin J. Lohse, Paolo Annibale, Roberto Maggio.

**Data curation:** Francesco Petragnano, Ilaria Pietrantoni, Paolo Annibale, Roberto Maggio.

**Formal analysis:** Francesco Petragnano, Franco Giorgi, Kostas Tokatlidis, Mario Rossi, Martin J. Lohse, Paolo Annibale, Roberto Maggio.

**Funding acquisition:** Martin J. Lohse, Paolo Annibale, Roberto Maggio.

**Investigation:** Irene Fasciani, Francesco Petragnano, Ziming Wang, Ruairidh Edwards, Narasimha Telugu, Ilaria Pietrantoni, Henrik Zauber, Marlies Grieben, Maria E. Terzenidou, Jacopo Di Gregorio, Cristina Pellegrini, Silvano Santini, Jr, Anna R. Taddei, Bärbel Pohl, Stefano Aringhieri, Marco Carli, Gabriella Aloisi, Eve Charlesworth, Alexandra Roman, Mario Rossi, Paolo Annibale, Roberto Maggio.

**Methodology:** Irene Fasciani, Francesco Petragnano, Narasimha Telugu, Ulrike Zabel, Henrik Zauber, Marlies Grieben, Bärbel Pohl, Paolo Annibale, Roberto Maggio.

**Project administration:** Paolo Annibale, Roberto Maggio.

**Resources:** Ulrike Zabel.

**Supervision:** Sebastian Diecke, Franco Giorgi, Fernanda Amicarelli, Andrew B. Tobin, Marco Scarselli, Kostas Tokatlidis, Martin J. Lohse, Paolo Annibale, Roberto Maggio.

**Visualization:** Irene Fasciani, Francesco Petragnano, Ilaria Pietrantoni, Paolo Annibale, Roberto Maggio.

**Writing – original draft:** Paolo Annibale, Roberto Maggio.

**Writing – review & editing:** Francesco Marampon, Vincenzo Flati, Marco Scarselli, Kostas Tokatlidis, Mario Rossi, Martin J. Lohse, Paolo Annibale, Roberto Maggio.

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
