## [Editor Report · Decision Letter 0]

29 Nov 2022

Dear Dr Annibale, 

Thank you for submitting your manuscript entitled "One gene - two proteins: The C-terminus of the prototypical M2 muscarinic receptor localizes to the mitochondria and regulates cell respiration" for consideration as a Research Article by PLOS Biology.

Your manuscript has now been evaluated by the PLOS Biology editorial staff as well as by an academic editor with relevant expertise and I am writing to let you know that we would like to send your submission out for external peer review.

Once your full submission is complete, your paper will undergo a series of checks in preparation for peer review. After your manuscript has passed the checks it will be sent out for review. To provide the metadata for your submission, please Login to Editorial Manager (https://www.editorialmanager.com/pbiology) within two working days, i.e. by Dec 01 2022 11:59PM.

Kind regards,

Ines

--

Ines Alvarez-Garcia, PhD

Senior Editor

PLOS Biology

---

## [Decision Letter · Decision Letter 1]

27 Feb 2023

Dear Dr Annibale,

Thank you very much for your patience while your manuscript entitled "One gene - two proteins: The C-terminus of the prototypical M2 muscarinic receptor localizes to the mitochondria and regulates cell respiration" was peer-reviewed at PLOS Biology. Also, please accept my sincere apologies for the long delay in sending you our decision. Your study has now been assessed and discussed by the PLOS Biology editors, an Academic Editor with relevant expertise and two independent reviewers.

The reviews are attached below. As you will see, both reviewers find the conclusions potentially interesting, however they also raise significant concerns that would need to be addressed in order to confirm the results of the study. Reviewer 1 thinks that the descriptions of the figures are not accurate and mentions several missing controls that are crucial to confirm the findings, among other issues. Reviewer 2 has similar comments, missing several critical controls in the experiments, but also questions how often cap-independent translation of M2 occurs, if it is physiologically significant and how general the findings are regarding other GPCRs.

Based on these reviews and our discussions with the Academic Editor, I regret that we will not be able to invite revision of the study, as successfully addressing the concerns raised would entail a significant amount of work (beyond what could be reasonably expected of a revision) and whether the current conclusions would be supported is unclear. However, given our and the reviewers' interest in the topic, if further work were to allow you to comprehensively address all the reviewers' concerns, we would be willing to consider an extensively revised manuscript as a re-submission. The article would receive a new number and submission date and we would evaluate it in view of any related work published in the interim, but we would make every attempt to engage the same Academic Editor and reviewers in its assessment, if no novelty concerns arise.

We appreciate that the scale of the requested additional work is significant and understand that you may prefer to pursue faster publication of this work elsewhere. At this point, your manuscript is no longer under active consideration at PLOS Biology. We might be able to facilitate a faster consideration at another PLOS journal; if this is of interest to you, please let me know.

Please feel free to contact me if you would like to discuss next steps for your manuscript. If you do make the choice to revise and re-submit to PLOS Biology, please provide a point-by-point response to the reviewers' comments when re-submitting. You should include this as a separate file, labelled 'Response to Reviewers'. Please also provide the tracking number of this manuscript [PBIOLOGY-D-22-02557R1] in the 'previous interaction' section of the online submission form and in your cover letter.

I am sorry that we cannot be more positive on this occasion. I thank you for having considered PLOS Biology for publication and hope that you will find the reviewers' feedback helpful as you consider how to proceed with this work.

Sincerely,

Ines

--

Ines Alvarez-Garcia, PhD

Senior Editor

PLOS Biology

Reviewers' comments

Rev. 1:

This study by Fasciani & Petragnano et al, is original and of great interest in the field of G protein coupled receptors (GPCRs). The study reveals a cap-independent translation of the C-terminal part (Ctail) of the M2 muscarinic receptor, involving an internal ribosomal entry site (IRES) at the level of the DNA sequence corresponding to the internal loop i3 of the receptor. The authors use different strategies to reveal this IRES-dependent expression of the Ctail and show that the expression of the Ctail fragment is increased under starvation conditions of the culture medium and that it localizes to mitochondria. Under stressed conditions induced by cellular starvation, the Ctail interacts with a component of the respiratory chain in mitochondria to decrease mitochondrial respiration, decrease ROS production and decrease cell proliferation. The authors conduct their study mainly in a HEK293 cell model but also extend it to mouse tissues and human iPSC cells where they show the existence of the M2 Ctail fragment and the impact of its expression. This study deserves to be published in Plos Biology. However, some controls need to be added, some corrections need to be made to improve the clarity of the statements, and some concerns and interpretations need to be clarified.

1. It is difficult to understand and analyse the figures when the descriptions and text are not exact, or sometimes impossible to read. Authors are strongly encouraged to proofread the manuscript and correct inaccuracies

- Label of fig1A is missing, is the second panel M2-STOP228-EGFP ?

We do not see cells in the insert (brightfield) of fig1A of non transfected

- Fig2 is difficult to understand

Annotations are of very bad quality and impossible to read

Fig2A and 2C text is not of good quality, it is very difficult to read the text. Idem for Fig3, the text is not visible

- Lane 240 "This is as also confirmed from ribosome profiling data from human left ventricle tissue, displaying increased p-sites as well as overall ribosomal coverage in correspondence of this region (Fig. S4B)". what are "p-sites" ? Figure S2B is just impossible to read

- Graph Fig4B: impossible to read the Y axis legend

- FigS8 legends: S8A "below" is not the lower panel but the right panel

- Fig S10B-S10D legends are too small- Impossible to read

2. Controls are needed

2.1. Lane 122-FigS2A-Table1: To fully validate the hypothesis of a reconstitution of competent M2 receptor with the Nter and Cter parts of the recepror when a stop codon is introduced at position 228 of M2, coexpression of the Nter fragment (1-228, comprising only TM1-5 trunk, without the rest in the DNA construct) with the Cter tail (228-446, comprising only TM6-7) would be necessary. Indeed the residual binding observed with M2-Stop228 could be due to expression of a weak fraction of full length M2 despite the introduction of a stop at position 228 since ribosomes can have the properties to go through and pass STOP codon. Same remark for M3 in Table S2 and figS5. Furthermore, Western Blot profiling of the different constructs should also be shown

2.2. TableS1: Explain why the coexpression of MT2-STOP196 + M2-Trunk1-283 shows some binding; properties it is rather unexpected. Furthermore, controls of expression of all the constructs must be provided; without checking level of expression of mutant constructs, conclusions can be erroneous. Same remarks for the experiments related to the M3 receptor

2.3 Fig3C: The authors should show the expression of each construct in the total lysate compared to the mitochondrial fraction. Evaluation of expression via IRES vs cap dependent expression would be interesting to know.

2.4. Fig 4C in vitro import into yeast mitochondria: I have the feeling that some controls are lacking: for ex, same condition of expression and import but this time in the absence of yeast mitochondria ?

3. Inaccuracies and concerns:

- Fig3B: M2 is not well visible at the cell surface in 1B in contrast to what is seen in other figures

- Fig3E: mitochondrial localization is not possible to see clearly, the images are not convincing. In addition, separate spectral green channel displayed at the bottom does not seem to correspond to the one from the merge top image

- FigS6J: The WB dispalys several bands (of which a predominant one below 17kDa and several above 55kDa). What are all the bands seen in WB ? What is the size of Ful length M2 ? what is the size of C-tail fragment? This should be specified in the legend and in the text

- Lane 340 and FigS7: The authors wrote "Similar mitochondrial localization was also observed when we transfected HEK293 cells with M2fr.sh-mRuby2 (Fig. S7B)." However in FigS7B, mitotracker staining is missing.

- Fig4A images are of very poor quality - impossible to judge the integration into mitochondria

- FigS8:

FigS8: scales are indicated to be 10um; however the scale bar size is different in all images but cell size seem to be similar….

There is no nuclear staining here, for some images, it would be necessary to include it

Idem nuclear staining of S6 would help. Plasma membrane is difficult to distinguish

Fig S8A: Indicate the MW of M2 Ctail incorporated into yeast mitochondria ? Is it the one <17kDa

Fig S8E and S8G: For the right interpretation, controls of expression of the constructs should be shown

- Lane 442 and Fig5C-5D: "The green fluorescence detected at the plasma membrane is explained with the chaperoning effect of M2-GFP11 on Mito-GFP1-10 before its sorting to the mitochondria". This observation calls into question the Mito-GFP1-10 sensor. Indeed Mito-GFP1-10 being observed at the plasma membrane instead of mitochondria suggests that the mitochondrial sensor is not specific; What are the comments of the authors regarding this aspect ?

- The physiological role of Ctail is not clear, its impact on cellular apoptosis or proliferation is not clear. Indeed:

FigS10D: Expression of M2stop228, M2Stop400 and M2tail increases cleaved caspase 3 compared to WT, suggesting that expression of the 3 constructs including C-tail induced some cell death (while WT does not). However, the authors wrote in the text Lane 483: "no difference in the number of apoptotic cells among the different cell groups as shown by the caspase-3 activation assays of apoptosis" Is there a discrepancy here ? Does the Ctail increase apoptosis ?

Ctail-induced apoptosis would also match the increased cell viability assay on HEK293 cells transfected with M2-M368A (ie in cells not expressing Ctail) (Fig6), or the cell proliferation assay performed on M368A hiPSCs (Fig7D), compared to WT receptor

Extra experiments would certainly be needed to clarify this aspect

Minor:

Lane 230: Is the word "However" appropriate here ?

Lane 240, "This is as also confirmed… » remove "as"

Lane 328 A word is missing at the end of the sentence: " ….the mechanism of C-terminal fragment production is therefore close to physiological."

Fig3D legends, remove « as well as strongly to the cell mitochondria »; since nothing in 3D show that the localisation is in mitochondria; replace by intracellular compartments or organelles. Idem in the text lane 332

-Fig2

"OF" is missing in the Title of fig2: "Localization of the IRES within the i3 loop OF the muscarinic M 209 2 receptor using a fluorescence reporter assay."

The different constructs of fig2A are not easy to understand; adding colors for the different parts of the i3 loop fragments would help the reader

- Figure 4

Fig 4C in vitro import into yeast mitochondria:

Legend does not seem correct. It is said line 428 « The kinetics pattern of M2tail is what it is expected for a protein imported in mitochondria: the appearance of the cleaved M2-tail is indicative of a post translational modifications of the fragment once imported in mitochondria.» I do not see the notion of post-translational modification here; what do the authors mean ?

What is the line at 5% correspond to ? Nothing is indicated in the legend.

Fig4E. What is TEM ?

Fig4F-G: specify in the legend the antibody associated with gold particles, and also needs controls with cells without M2

-Figure S9

FigS9D� colocalisation should be dispalyed in white

Need zoom

FigS9H � mitochondria (mitotracker) are barely visible

Again scale of the images on top and bottom are not the same

- Figure S10

Put a distinguishable box to differentiate the top and bottom panels of S10D

Put a box to distinguish the 3 panels of S10E

Rev. 2:

In this paper, the authors reported that there is an internal ribosome entry site (IRES) in the third intracellular loop of M2 receptor which splits M2 receptor into two fragments. They showed that the carboxyl terminal fragment locates to the mitochondria inner membrane and regulates various functions including oxygen consumption and cell proliferation. The authors employed various techniques and a lot of different constructs to comprehensively map out the IRES site, report the subcellular localization and assess the functions. I think this is interesting and potentially impactful, but I have a few comments as detailed below.

1. How often does cap-independent translation of M2 happen and does it have significant impact in physiological conditions? Majority of M2 receptor locates at the plasma membrane as shown by the authors themselves and many other papers. If less than 5% locates to the mitochondria, would that have any significant impact? The authors tried to address this question using the M2-tdtomato knock-in mice and use western blot to look at the expression of M2-tdtomato and M2-C tail in the tissue and purified mitochondria. These bands are not quantifiable and I think it is of great importance to report the approximate ratio of M2 C tail located in mitochondria out of the total M2. Moreover, the size of the band is questionable. The M2 receptor has a molecular weight around 52 kDa, not the 75kDa as stated in the paper (Line 558). This also directly contradicts with authors' own western blot in Figure 4C where the M2 runs a bit lower than 55 kDa. This raised the question whether the bands in Fig. 7A are really the M2 receptor bands. All the controls are without marker and the quality could be improved.

2. Generalization to other members of the GPCR superfamily? The authors could only identify M2 and M3 out of more than 800 different GPCRs. M2 and M3 has a very long i3 loop which are not common in GPCR family, so could the long i3 loop needed for cap-independent translation? so that it only limits to few GPCRs with long i3 loop?

3. Fig. 3B the M2 receptor localizes in the cytosol instead of the plasma membrane, which is not canonical localization as the authors claimed. In fact, the authors contradict themselves in the very same figure. In figure 2D, the M2-mRuby2 shows a predominant cell plasma membrane localization.

4. Fig, 3E. there are a large amount of green signal which overlaps with mitotracker, but there are also large amount of green signal which are not. Those green signals look like that they are not diffused. Where do they locate?

5. In Figure 4 E, F, G. I could see many similar particles in the images. Are these not real signals? Please clarify.

Minor:

Fig. 1A. top right panel add text M2stop228-EGFP.

Fig. 1A. Are they HEK293 or COS7 cells? "Imaging setting are the same for the five images" five or four? there are only four images in figure 1A.

Page 15, line 418, over a 4h interval? Do you mean 1h interval?

---

## [Editor Report · Decision Letter 2]

20 Apr 2023

Dear Dr Annibale,

As we mentioned in our previous letter, we have now changed the decision of your manuscript entitled "One gene - two proteins: The C-terminus of the prototypical M2 muscarinic receptor localizes to the mitochondria and regulates cell respiration" and you will be able to submit a revised version of the manuscript.

Please address all the comments of the reviewers to support the conclusions of the manuscript. Given the extent of revision needed, we cannot make a decision about publication until we have seen the revised manuscript and your response to the reviewers' comments. Your revised manuscript is likely to be sent for further evaluation by all or a subset of the reviewers.

**IMPORTANT - SUBMITTING YOUR REVISION**

3. Resubmission Checklist

a) *PLOS Data Policy*

b) *Published Peer Review*

Sincerely,

Ines

--

Ines Alvarez-Garcia, PhD

Senior Editor

PLOS Biology

Reviewers' comments

Rev. 1:

This study by Fasciani & Petragnano et al, is original and of great interest in the field of G protein coupled receptors (GPCRs). The study reveals a cap-independent translation of the C-terminal part (Ctail) of the M2 muscarinic receptor, involving an internal ribosomal entry site (IRES) at the level of the DNA sequence corresponding to the internal loop i3 of the receptor. The authors use different strategies to reveal this IRES-dependent expression of the Ctail and show that the expression of the Ctail fragment is increased under starvation conditions of the culture medium and that it localizes to mitochondria. Under stressed conditions induced by cellular starvation, the Ctail interacts with a component of the respiratory chain in mitochondria to decrease mitochondrial respiration, decrease ROS production and decrease cell proliferation. The authors conduct their study mainly in a HEK293 cell model but also extend it to mouse tissues and human iPSC cells where they show the existence of the M2 Ctail fragment and the impact of its expression. This study deserves to be published in Plos Biology. However, some controls need to be added, some corrections need to be made to improve the clarity of the statements, and some concerns and interpretations need to be clarified.

1. It is difficult to understand and analyse the figures when the descriptions and text are not exact, or sometimes impossible to read. Authors are strongly encouraged to proofread the manuscript and correct inaccuracies

- Label of fig1A is missing, is the second panel M2-STOP228-EGFP ?

We do not see cells in the insert (brightfield) of fig1A of non transfected

- Fig2 is difficult to understand

Annotations are of very bad quality and impossible to read

Fig2A and 2C text is not of good quality, it is very difficult to read the text. Idem for Fig3, the text is not visible

- Lane 240 "This is as also confirmed from ribosome profiling data from human left ventricle tissue, displaying increased p-sites as well as overall ribosomal coverage in correspondence of this region (Fig. S4B)". what are "p-sites" ? Figure S2B is just impossible to read

- Graph Fig4B: impossible to read the Y axis legend

- FigS8 legends: S8A "below" is not the lower panel but the right panel

- Fig S10B-S10D legends are too small- Impossible to read

2. Controls are needed

2.1. Lane 122-FigS2A-Table1: To fully validate the hypothesis of a reconstitution of competent M2 receptor with the Nter and Cter parts of the recepror when a stop codon is introduced at position 228 of M2, coexpression of the Nter fragment (1-228, comprising only TM1-5 trunk, without the rest in the DNA construct) with the Cter tail (228-446, comprising only TM6-7) would be necessary. Indeed the residual binding observed with M2-Stop228 could be due to expression of a weak fraction of full length M2 despite the introduction of a stop at position 228 since ribosomes can have the properties to go through and pass STOP codon. Same remark for M3 in Table S2 and figS5. Furthermore, Western Blot profiling of the different constructs should also be shown

2.2. TableS1: Explain why the coexpression of MT2-STOP196 + M2-Trunk1-283 shows some binding; properties it is rather unexpected. Furthermore, controls of expression of all the constructs must be provided; without checking level of expression of mutant constructs, conclusions can be erroneous. Same remarks for the experiments related to the M3 receptor

2.3 Fig3C: The authors should show the expression of each construct in the total lysate compared to the mitochondrial fraction. Evaluation of expression via IRES vs cap dependent expression would be interesting to know.

2.4. Fig 4C in vitro import into yeast mitochondria: I have the feeling that some controls are lacking: for ex, same condition of expression and import but this time in the absence of yeast mitochondria ?

3. Inaccuracies and concerns:

- Fig3B: M2 is not well visible at the cell surface in 1B in contrast to what is seen in other figures

- Fig3E: mitochondrial localization is not possible to see clearly, the images are not convincing. In addition, separate spectral green channel displayed at the bottom does not seem to correspond to the one from the merge top image

- FigS6J: The WB dispalys several bands (of which a predominant one below 17kDa and several above 55kDa). What are all the bands seen in WB ? What is the size of Ful length M2 ? what is the size of C-tail fragment? This should be specified in the legend and in the text

- Lane 340 and FigS7: The authors wrote "Similar mitochondrial localization was also observed when we transfected HEK293 cells with M2fr.sh-mRuby2 (Fig. S7B)." However in FigS7B, mitotracker staining is missing.

- Fig4A images are of very poor quality - impossible to judge the integration into mitochondria

- FigS8:

FigS8: scales are indicated to be 10um; however the scale bar size is different in all images but cell size seem to be similar….

There is no nuclear staining here, for some images, it would be necessary to include it

Idem nuclear staining of S6 would help. Plasma membrane is difficult to distinguish

Fig S8A: Indicate the MW of M2 Ctail incorporated into yeast mitochondria ? Is it the one <17kDa

Fig S8E and S8G: For the right interpretation, controls of expression of the constructs should be shown

- Lane 442 and Fig5C-5D: "The green fluorescence detected at the plasma membrane is explained with the chaperoning effect of M2-GFP11 on Mito-GFP1-10 before its sorting to the mitochondria". This observation calls into question the Mito-GFP1-10 sensor. Indeed Mito-GFP1-10 being observed at the plasma membrane instead of mitochondria suggests that the mitochondrial sensor is not specific; What are the comments of the authors regarding this aspect ?

- The physiological role of Ctail is not clear, its impact on cellular apoptosis or proliferation is not clear. Indeed:

FigS10D: Expression of M2stop228, M2Stop400 and M2tail increases cleaved caspase 3 compared to WT, suggesting that expression of the 3 constructs including C-tail induced some cell death (while WT does not). However, the authors wrote in the text Lane 483: "no difference in the number of apoptotic cells among the different cell groups as shown by the caspase-3 activation assays of apoptosis" Is there a discrepancy here ? Does the Ctail increase apoptosis ?

Ctail-induced apoptosis would also match the increased cell viability assay on HEK293 cells transfected with M2-M368A (ie in cells not expressing Ctail) (Fig6), or the cell proliferation assay performed on M368A hiPSCs (Fig7D), compared to WT receptor

Extra experiments would certainly be needed to clarify this aspect

Minor:

Lane 230: Is the word "However" appropriate here ?

Lane 240, "This is as also confirmed… » remove "as"

Lane 328 A word is missing at the end of the sentence: " ….the mechanism of C-terminal fragment production is therefore close to physiological."

Fig3D legends, remove « as well as strongly to the cell mitochondria »; since nothing in 3D show that the localisation is in mitochondria; replace by intracellular compartments or organelles. Idem in the text lane 332

-Fig2

"OF" is missing in the Title of fig2: "Localization of the IRES within the i3 loop OF the muscarinic M 209 2 receptor using a fluorescence reporter assay."

The different constructs of fig2A are not easy to understand; adding colors for the different parts of the i3 loop fragments would help the reader

- Figure 4

Fig 4C in vitro import into yeast mitochondria:

Legend does not seem correct. It is said line 428 « The kinetics pattern of M2tail is what it is expected for a protein imported in mitochondria: the appearance of the cleaved M2-tail is indicative of a post translational modifications of the fragment once imported in mitochondria.» I do not see the notion of post-translational modification here; what do the authors mean ?

What is the line at 5% correspond to ? Nothing is indicated in the legend.

Fig4E. What is TEM ?

Fig4F-G: specify in the legend the antibody associated with gold particles, and also needs controls with cells without M2

-Figure S9

FigS9D� colocalisation should be dispalyed in white

Need zoom

FigS9H � mitochondria (mitotracker) are barely visible

Again scale of the images on top and bottom are not the same

- Figure S10

Put a distinguishable box to differentiate the top and bottom panels of S10D

Put a box to distinguish the 3 panels of S10E

Rev. 2:

In this paper, the authors reported that there is an internal ribosome entry site (IRES) in the third intracellular loop of M2 receptor which splits M2 receptor into two fragments. They showed that the carboxyl terminal fragment locates to the mitochondria inner membrane and regulates various functions including oxygen consumption and cell proliferation. The authors employed various techniques and a lot of different constructs to comprehensively map out the IRES site, report the subcellular localization and assess the functions. I think this is interesting and potentially impactful, but I have a few comments as detailed below.

1. How often does cap-independent translation of M2 happen and does it have significant impact in physiological conditions? Majority of M2 receptor locates at the plasma membrane as shown by the authors themselves and many other papers. If less than 5% locates to the mitochondria, would that have any significant impact? The authors tried to address this question using the M2-tdtomato knock-in mice and use western blot to look at the expression of M2-tdtomato and M2-C tail in the tissue and purified mitochondria. These bands are not quantifiable and I think it is of great importance to report the approximate ratio of M2 C tail located in mitochondria out of the total M2. Moreover, the size of the band is questionable. The M2 receptor has a molecular weight around 52 kDa, not the 75kDa as stated in the paper (Line 558). This also directly contradicts with authors' own western blot in Figure 4C where the M2 runs a bit lower than 55 kDa. This raised the question whether the bands in Fig. 7A are really the M2 receptor bands. All the controls are without marker and the quality could be improved.

2. Generalization to other members of the GPCR superfamily? The authors could only identify M2 and M3 out of more than 800 different GPCRs. M2 and M3 has a very long i3 loop which are not common in GPCR family, so could the long i3 loop needed for cap-independent translation? so that it only limits to few GPCRs with long i3 loop?

3. Fig. 3B the M2 receptor localizes in the cytosol instead of the plasma membrane, which is not canonical localization as the authors claimed. In fact, the authors contradict themselves in the very same figure. In figure 2D, the M2-mRuby2 shows a predominant cell plasma membrane localization.

4. Fig, 3E. there are a large amount of green signal which overlaps with mitotracker, but there are also large amount of green signal which are not. Those green signals look like that they are not diffused. Where

---

## [Decision Letter · Decision Letter 3]

9 Sep 2023

Dear Dr Annibale,

Thank you for your patience while we considered your revised manuscript entitled "One gene - two proteins: The C-terminus of the prototypical M2 muscarinic receptor localizes to the mitochondria and regulates cell respiration" for consideration as a Research Article at PLOS Biology. Your revised study has now been evaluated by the PLOS Biology editors, the Academic Editor and one of the original reviewers. 

The reviews and comments from the Academic Editor are attached below. While we were unable to obtain new comments from Reviewer 1, the Academic Editor has looked at your responses and thinks that the manuscript needs further improvements in order for us to consider it for publication. S/he thinks that you should incorporate better the data provided to the reviewers in the manuscript and add appropriate controls for all the westerns in a clear and logical manner. Reviewer 2 also raises a point that should be addressed.

In light of the reviews and the Academic Editor's comments, we have decided offer you the opportunity to address the remaining concerns in a revision. We will then assess your revised manuscript and your response to the reviewers' comments with our Academic Editor aiming to avoid further rounds of peer-review, although might need to consult with the reviewers, depending on the nature of the revisions.

We expect to receive your revised manuscript within 2 months. Please email us (plosbiology@plos.org) if you have any questions or concerns, or would like to request an extension. 

**IMPORTANT - SUBMITTING YOUR REVISION**

3. Resubmission Checklist

a) *PLOS Data Policy*

b) *Published Peer Review*

Sincerely,

Ines

--

Ines Alvarez-Garcia, PhD

Senior Editor

PLOS Biology

Reviewers' comments

Rev. 2:

The reviewers address the comments reasonably well. I appreciate that the authors inserted the figures into the text which makes the resolution so much better. That alone makes the paper easier to understand. 

One comment:

M2R molecular weight, I wouldn't call the insect cell expressed M2R "naked" because they are also glycosylated, but the pattern could be different that from the HEK cell. It could be better that the authors provide the references for that the M2R run at 72kDa. A quick literature search reveals mixed results, some papers show 52 kDa and some show ~72 kDa. I am not 100% convinced that glycosylation can cause a difference as big as 20 kDa in mobility shift.

Academic Editor's comments

I still find the manuscript difficult to follow. Both reviewers raised legitimate concerns and while the authors made a real effort to address these, only minor changes were made in the manuscript. Furthermore, many concerns raised by the reviewers regarding controls are not clearly addressed in the rebuttal, making it very difficult to follow the logic. As presented, most of the westerns do not have appropriate controls, or the controls are scattered in different blots and figure panels. Therefore, to meet the standards of PLOS Biology, the authors need to provide a significantly revised version of their manuscript that not only incorporates appropriate controls for each experiment, but also includes the data they provided to the reviewers in a clear and logical manner.

---

## [Editor Report · Decision Letter 4]

19 Dec 2023

Dear Dr Annibale,

Thanks again for your patience while we considered the new version of your manuscript entitled "One gene - two proteins: The C-terminus of the prototypical M2 muscarinic receptor localizes to the mitochondria and regulates cell respiration" as a Research Article at PLOS Biology. Your revised study has now been evaluated by the PLOS Biology editors and the Academic Editor. 

While the Academic Editor acknowledges the improvements made in the manuscript, still thinks that is difficult to read and follow, and that the manner in which a lot of the data is presented needs attention. For example, for most of the data it is not visible how many repeats are done and for the IF images and co-localization, there is no quantification included. We think it is very important to include them because the localization of the M2 tail to the mitochondria is key for the manuscript. Furthermore, phase and merged images are sometimes missing - please make sure these are added in all relevant figures. You will also need to provide more information regarding the actual molecular weights of the various forms of the M2 receptor. As a guideline, you can find attached an annotated version of the manuscript highlighting some of the problems in the current version, however this is not a comprehensive review and the manner by which the data is presented needs to be carefully reviewed before we can make a final editorial decision.

**IMPORTANT - SUBMITTING YOUR REVISION**

Please submit the following files along with your revised manuscript:

Please don't hesitate to contact us if you have any questions or comments.

Sincerely,

Ines

--

Ines Alvarez-Garcia, PhD

Senior Editor

PLOS Biology

---

## [Editor Report · Decision Letter 5]

9 Feb 2024

Dear Dr Annibale,

Thank you for your patience while we considered your revised manuscript entitled "One gene - two proteins: The C-terminus of the prototypical M2 muscarinic receptor localizes to the mitochondria and regulates cell respiration." for publication as a Research Article at PLOS Biology. This revised version of your manuscript has been evaluated by the PLOS Biology editors and by the Academic Editor.

Based on our Academic Editor's assessment of your revision, we are likely to accept this manuscript for publication, provided you satisfactorily address the data and other policy-related requests stated below.

In addition, we would like you to consider a suggestion to improve the title:

"The C-terminus of the prototypical M2 muscarinic receptor localizes to the mitochondria and regulates cell respiration under stress conditions"

We expect to receive your revised manuscript within two weeks. 

*Published Peer Review History*

*Press*

Sincerely,

Ines

--

Ines Alvarez-Garcia, PhD

Senior Editor

PLOS Biology

ETHICS STATEMENT:

Given that your study includes experiments using mice, we do require an Ethics Statement. Please note this in the Metadata and also provide a license number in the statement you have included in the Methods section.

Fig. 2A, C; Fig. 4B; Fig. 5A, B; Fig. 6A-D; Fig. 7D, E; Fig. S2A; Fig. S5B; Fig. S6D, G; Fig. S7E and Fig. S10A-C, F-H

NOTE: the numerical data provided should include all replicates - we do require at least 3 of each experiment - AND the way in which the plotted mean and errors were derived (it should not present only the mean/average values).

*** Please also make the imaging data you have deposited on IDR publicly available.

SPECIES INDICATED IN THE ABSTRACT? 

- Please note that per journal policy, the model system/species studied should be clearly stated in the abstract of your manuscript. Please mention in the Abstract that you are using yeast and mice in the study.

Many thanks for providing a file that includes the original raw images of the gels. I have checked them and we are missing the gels shown in the following figures, thus please provide them and update the file:

Fig. 4C; Fig. S2A, B; Fig. S7D and Fig. S10D

---

## [Editor Report · Decision Letter 6]

11 Mar 2024

Dear Dr Annibale,

Thank you for the submission of your revised Research Article entitled "The C-terminus of the prototypical M2 muscarinic receptor localizes to the mitochondria and regulates cell respiration under stress conditions" for publication in PLOS Biology. On behalf of my colleagues and the Academic Editor, Carole Parent, I am delighted to let you know that we can in principle accept your manuscript for publication, provided you address any remaining formatting and reporting issues. These will be detailed in an email you should receive within 2-3 business days from our colleagues in the journal operations team; no action is required from you until then. Please note that we will not be able to formally accept your manuscript and schedule it for publication until you have completed any requested changes.

PRESS

Sincerely, 

Ines

--

Ines Alvarez-Garcia, PhD

Senior Editor

PLOS Biology
